# Structural defects in amyloid-β fibrils drive secondary nucleation

Jing Hu [1,2], Tom Scheidt [3,4], Dev Thacker [2,5], Emil Axell [2], Elin Stemme[2], Urszula Łapińska [6], Stefan Wennmalm[7], Georg Meisl [3,8], Samo Curk[9], Maria Andreasen [10], Michele Vendruscolo [3], Paolo Arosio [11], Anđela Šarić [9], Jeremy D. Schmit[12], Tuomas P. J. Knowles [3,13], Emma Sparr [1], Sara Linse [2], Thomas C. T. Michaels [14,15] & Alexander J. Dear [2,14,15] ✉

Formation of new amyloid fibrils and oligomers from monomeric protein on the surfaces of existing fibrils is an important driver of many disorders such as Alzheimer's and Parkinson's diseases. The structural basis of this secondary nucleation process, however, is poorly understood. Here, we ask whether secondary nucleation sites are found predominantly at rare growth defects: irregularities in the fibril core structure incorporated during their original assembly. We first demonstrate using the specific inhibitor of secondary nucleation, Brichos, that secondary nucleation sites on Alzheimer's disease-associated fibrils composed of Aβ40 and Aβ42 peptides are rare compared to the number of protein molecules they contain. We then grow Aβ40 fibrils under conditions designed to eliminate most growth defects while leaving the regular fibril morphology unchanged, and confirm the latter using cryo-electron microscopy. We measure both the ability of these annealed fibrils to promote secondary nucleation and the stoichiometry of their secondary nucleation sites, finding that both are greatly reduced as predicted. Re-analysis of published data for other proteins suggests that fibril growth defects may also drive secondary nucleation generally across most amyloids. These findings could unlock structure-based drug design of therapeutics that aim to halt amyloid disorders by inhibiting secondary nucleation sites.

Amyloid fibrils are characterised by the stacking of proteins into β-sheet-rich aggregates with a hydrophobic core. They can serve functional purposes but are also frequently associated with pathology[1–4], making the mechanisms that control their proliferation a central problem in disease. Their spontaneous formation requires a slow primary nucleation process producing very short new fibrils from monomeric protein, and their subsequent rapid elongation by monomer addition (Fig. 1a)[5]. Notably, in many systems, an additional reaction step, secondary nucleation of new fibrils on the surfaces of existing fibrils[6–9], dominates new oligomer and fibril production once aggregates appear[10,11]. Secondary nucleation has been implicated across systems including type II diabetes-associated islet amyloid polypeptide

(IAPP)[12,13], Parkinson's disease-associated α-synuclein[14], and Alzheimer's disease (AD)-associated tau and amyloid-β (Aβ)[10,15,16]. Rapid secondary nucleation is characteristic of many disease-associated amyloids, due both to its auto-catalytic effect on fibril proliferation and to its ability to produce large quantities of toxic protein oligomers[9,17]. Analogous surface-catalysed nucleation processes are well known in crystallisation, underscoring the generality of the underlying physics[18–20].

However, despite its importance, exactly how fibril surfaces catalyse secondary nucleation is still not known. A common assumption is that catalysis is uniform across the entire fibril surface, or at least that catalytic sites occur at every plane in the fibril (see Fig. 1b(i)).

Nonetheless, several recent studies imply secondary nucleation sites on Aβ fibrils are rare and substoichiometric[21–24], casting doubt on this hypothesis, although the stoichiometry of these sites has never been quantified directly. One proposed explanation for this is that catalysis is localised at rare super-spreader fibrils (fibrils with dense secondary nucleation sites), with the remaining majority of fibrils being incapable of secondary nucleation (see Fig. 1b(ii)). However, recent studies indicate that secondary nucleation sites are spatially isolated from one another[25], that most Aβ42 fibrils have comparable catalytic activity[26], and that almost all Aβ42 fibrils possess non-periodic secondary nucleation sites[27], rendering this model unlikely for Aβ fibrils.

In light of the above findings, it has been proposed that these sites instead correspond to growth defects: defects generated in the fibril core during its initial assembly[25]. Those types of defect structures that permit the next fibril plane to be assembled with the correct structure on top of the defective fibril plane without imposing an insurmountable free energy penalty should exist throughout the fibril length coexisting with otherwise uniform fibril morphology. One example of such a defect is when the monomers in a new fibril plane bind correctly with respect to one another and with the correct fold but laterally offset and potentially also out-of-register from the monomers in the preceding fibril plane during fibril elongation[28,29]. The next fibril plane can then be assembled in perfect alignment with the offset plane, allowing correct growth to resume (see Fig. 1b(iii)). Small dislocations only partially exposing a filament's cross-section are possible, as are larger dislocations that lead to entire filaments being misaligned, or even to fibril branching. Other plausible defect structures include (but are not limited to) fibril planes in which one or more of the monomers is only partially correctly folded, exposing part of the fibril's hydrophobic core.

Near thermodynamic equilibrium, the stoichiometry of growth defects (the number of defects per monomer in the fibril) should follow the Boltzmann distribution, becoming the exponential of the free energy penalty for forming a given type of defect[30]. Because fibrils are highly stable these penalties are expected to be large, leading to low defect incorporation into growing fibrils. Typically, however, experiments use high initial supersaturation (i.e. monomer concentration) to obtain sufficient fibril yield and reaction speed, driving fibril growth far from equilibrium. The average time between successive monomer binding events becomes much smaller than the average time taken for an incorrectly bound monomer to detach[29]. Therefore, many fibril planes are bound sequentially on top of a defective fibril plane before it can detach or re-fold. The resultant very strong kinetic trapping can greatly increase defect stoichiometry[29,31]. Increasing defect incorporation with increasing supersaturation is a well-documented feature of crystallisation[32], of which amyloid formation is often considered a one-dimensional analogue[33]. Conversely, the common manufacturing technique of annealing can reduce defects in crystalline or pseudo-crystalline materials by using thermal treatment to remove kinetic trapping[34–36], suggesting an experimental lever to modulate growth defects.

Here, we quantify secondary nucleation site stoichiometries on Aβ42 and Aβ40 fibrils using BRICHOS, a ~100-residue mammalian chaperone that selectively binds to and blocks these sites[21,37,38], enabling a direct readout of site numbers per fibril mass. Inspired by annealing, we then use temperature control to grow Aβ40 fibrils under conditions designed to minimise growth defects whilst preserving their morphology as verified by high-resolution cryo-electron microscopy. Brichos binding measurements reveal that their secondary nucleation site stoichiometry is also greatly reduced, and kinetic assays demonstrate that the secondary nucleation rate of such annealed fibrils is reduced commensurately, causally linking defects to secondary nucleation sites. Together, these results identify growth defects as the predominant driver of secondary nucleation on Aβ fibrils, quantify their abundance, and suggest a general, defect-based mechanism for amyloid secondary nucleation supported by thermodynamic arguments and re-analysis of prior studies.

## Results

### Brichos binds Aβ42 amyloid fibrils tightly with low stoichiometry

We first characterised the interaction between the Brichos chaperone domain and Aβ42 fibrils using a microfluidic diffusional sizing (MDS) platform that was previously developed to quantify the interactions between biomolecules in solution under native conditions[39,40]. In brief, this technique utilises the diffusion profiles of molecules in solution collected at multiple positions as they flow through a microfluidics channel (Fig. 2a). From the analysis of these diffusion profiles using the advection-diffusion equation, it is possible to extract the distribution of diffusion coefficients. Brichos molecules that are bound to amyloid fibrils diffuse through the channel cross-section much more slowly than unbound Brichos molecules, exhibiting two distinct diffusion profiles that can be detected and separated mathematically[40]. Note, this separation becomes inaccurate when one species makes up <10% of the mixture.

Alexa-488 labelled proSP-C Brichos was titrated against 24 μM Aβ42 amyloid fibrils at 21 °C. Two different incubation procedures were followed: addition of Brichos to Aβ42 fibrils generated in the absence of chaperone; and incubation of Brichos with monomeric Aβ42 at 37 °C until essentially all monomers had converted into fibrils. In both procedures the mixtures were subsequently incubated for 48 h at 21 °C prior to evaluation of binding. The resulting binding curves are shown in Fig. 2b, c; the data obtained through the two procedures are closely similar. We then fit to these data the Langmuir adsorption isotherm[41] $\frac{[C_{bound}]}{[S_{tot}]} = \frac{[C_{free}]}{K_D + [C_{free}]}$, where $K_D$ is the dissociation constant, $[C_{bound}]$ and $[C_{free}]$ are the concentrations of the bound and free chaperone, respectively. $[S_{tot}]$ is the total concentration of binding sites available on an Aβ42 fibril surface, representing the maximum concentration of bound molecular chaperone. This procedure yielded $K_D = 314$ nM at 21 °C and $[S_{tot}] = 168$ nM. Dividing $[S_{tot}]$ by the total concentration of monomer equivalents of Aβ42 in the fibrils revealed the binding stoichiometry $s$ to be about one Brichos molecule per (24 μM)/(168 nM)=143 Aβ42 monomers (Fig. 2c), i.e. $s = 1/143$. Brichos is predominantly trimeric in solution[38], but dissociates into monomers when binding to Aβ fibril secondary nucleation sites[22,42]. This should not greatly affect the fitted stoichiometry, since each monomer is individually labelled and MDS measures total fluorescence. However, mild self-quenching of trimeric free Brichos is possible, in which case our fitted stoichiometry value would be a mild overestimate.

### Modulation of Aβ42 aggregation by Brichos reveals rare secondary nucleation sites

We next sought to relate our binding measurements to the ability of Brichos to inhibit secondary nucleation of Aβ42 fibrils. To this end, we incubated 3 μM of Aβ42 at 21 °C and monitored the time course of amyloid fibril formation by Thioflavin T (ThT) fluorescence in the absence and presence of increasing concentrations of Brichos (Fig. 2d). We observe that the addition of Brichos delays aggregation by an amount that depends on the concentration of the chaperone, in line with previous reports[21]. To disentangle the effect of Brichos on the rate constants for the individual microscopic steps of aggregation (primary nucleation $k_n$, secondary nucleation $k_2$ and elongation $k_+$), we compared the experimental kinetic profiles with a theoretical chemical kinetics model of aggregation in the presence of Brichos (see Methods "Kinetic modelling"). In this model, chaperone molecules can bind surface sites of fibrils and in this manner reduce the rate of secondary nucleation, while leaving the rate of the other aggregation steps unaffected[43]. A prediction from this model based on the unperturbed aggregation rate parameters obtained in the absence of Brichos (left-most curve in Fig. 2d) and the rate parameters determined from the

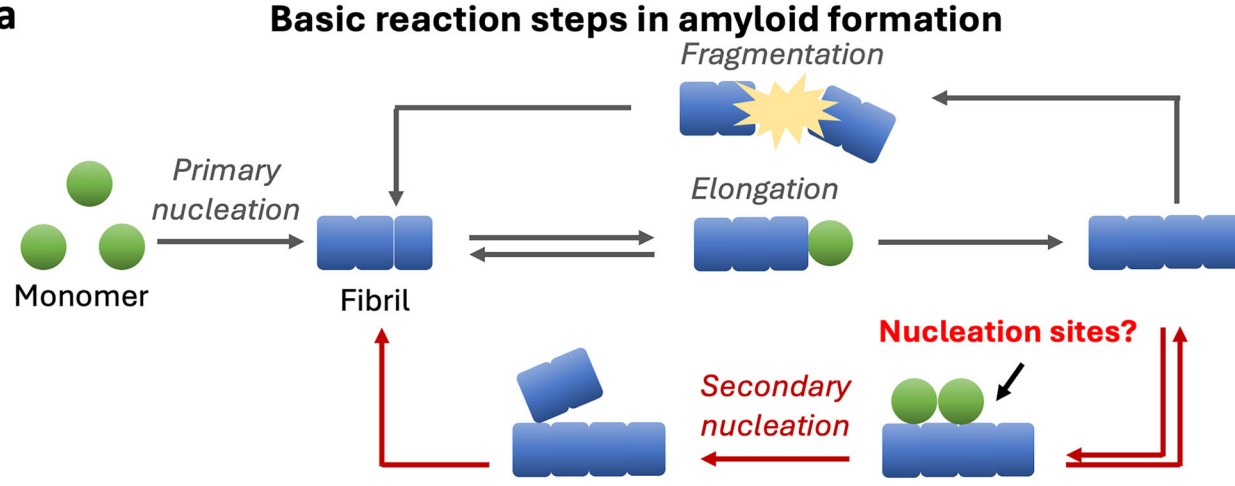

## a    Basic reaction steps in amyloid formation

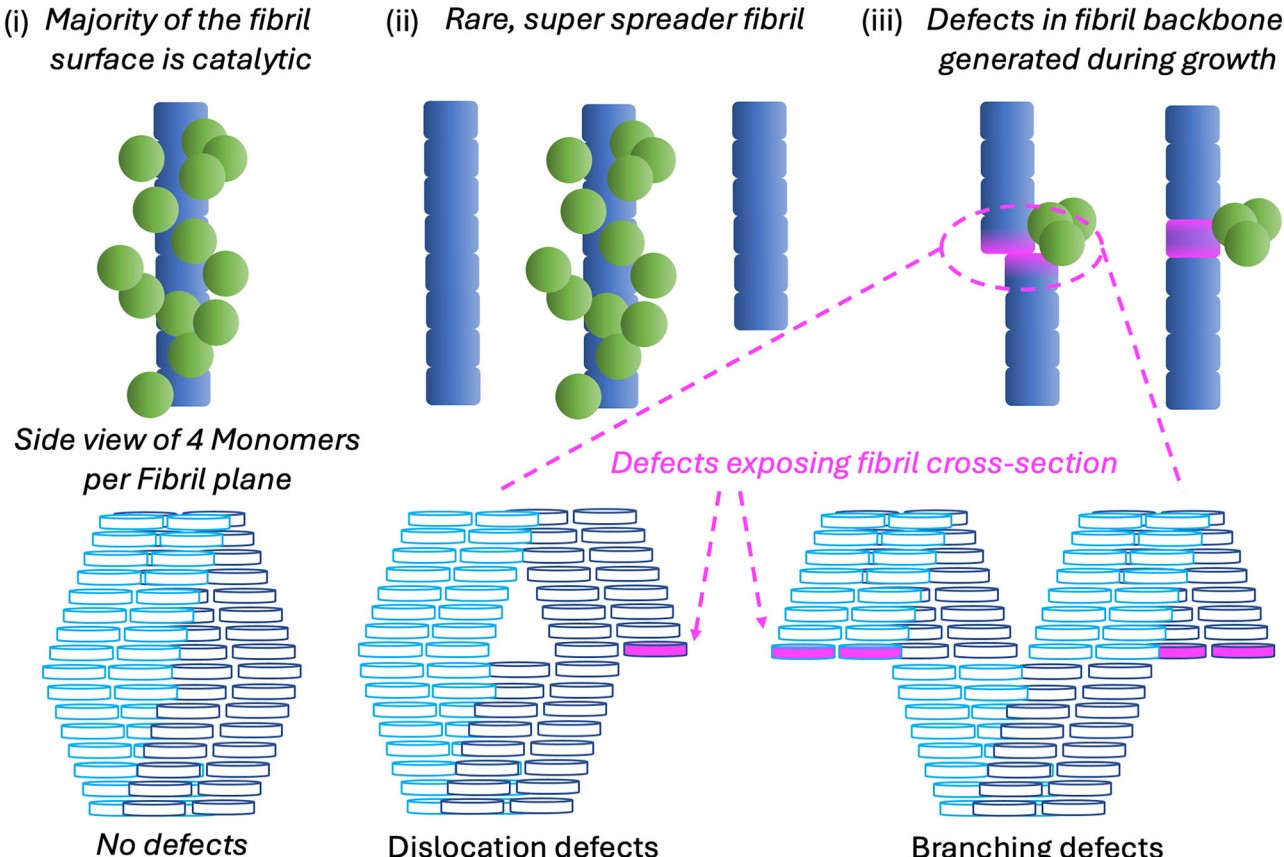

## b    Possible locations of secondary nucleation catalysis

(i) *Majority of the fibril surface is catalytic*

(ii) *Rare, super spreader fibril*

(iii) *Defects in fibril backbone generated during growth*

*Side view of 4 Monomers per Fibril plane*

*Defects exposing fibril cross-section*

*No defects*    Dislocation defects    Branching defects

**Fig. 1 | Secondary nucleation is a core process driving amyloid fibril proliferation; however, its structural basis is not understood. a** Overall reaction network governing amyloid fibril formation. Secondary nucleation is often responsible for most new fibril formation. **b** Possible ways amyloid fibrils might promote secondary nucleation of new fibrils (and oligomers). **(i):** Uniform catalysis of nucleation across the entire fibril surface. **(ii):** Rare super-spreader fibrils have dense catalytic sites, with most fibrils being incapable of promoting secondary nucleation. **(iii):** Secondary nucleation sites are defects in the fibril core, created during fibril elongation. Such defects can coincide with dislocations or branches (LHS, and bottom row), although non-offset defect structures are also possible (RHS). Dark and light blue colours distinguish separate protofilaments. Fibril cross-sections denoted by purple units and marked by arrows are exposed at dislocation or branching defect sites. The fraction of monomer units exposed may differ from that illustrated here. For simplicity, defects in subsequent figures will be represented schematically as dislocations only.

binding experiment shows good agreement with the experimental data. A comparison between the effective rate constant for secondary nucleation $k_2$ with increasing Brichos concentration and the binding curve of Brichos to fibrils confirms that the extent of inhibition of Aβ42 by Brichos closely correlates with the amount of chaperone bound to fibrils (Fig. 2e). Strikingly, we see that almost complete suppression of secondary nucleation (reduction of $k_2$ by >90%, see 2 and 3 μM Brichos datapoints) occurs with a Brichos coverage of <1% of the fibrils. We

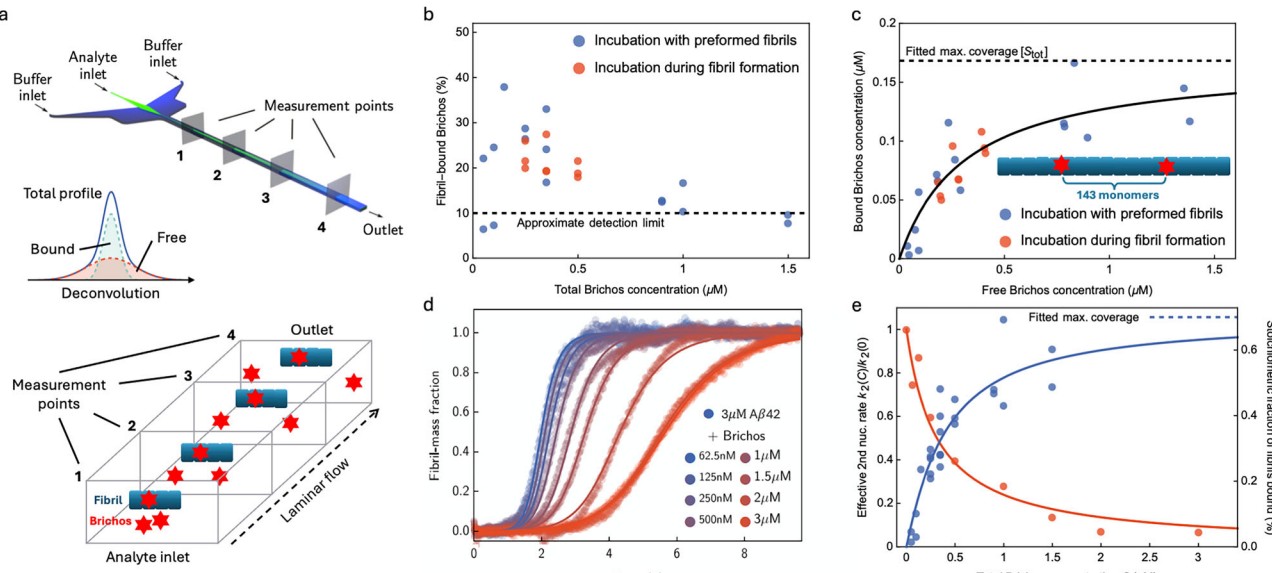

**Fig. 2 | Stoichiometry of Brichos binding to Aβ42 fibrils is low, and is predictive of secondary nucleation inhibition. a** Illustration of MDS experiment. Mixtures of Alexa-488-labelled proSP-C Brichos chaperone and 24 μM Aβ42 fibrils are introduced to the centre of a microfluidic channel. Free Brichos diffuses rapidly toward the channel edges, while fibril-bound Brichos displays slower lateral diffusion and remains therefore localised in the centre of the channel. Top schematic reprinted ("Adapted" or "in part") with permission from ref. 40. Copyright 2016 American Chemical Society. **b** The proportion of Brichos bound to fibrils, inferred by mathematical analysis of the diffusion profiles, decreases with increasing total concentration $C$ of proSP-C Brichos. By $C = 1.5$ μM it is already below the reliable detection limit of 10%; measurements at higher $C$ would therefore be uninformative. Sample sizes consisted of 2–3 repeats per condition. **c** Converting these proportions to free (x axis) and bound (y axis) Brichos concentrations enables global regression of a Langmuir adsorption isotherm to both datasets. This yields $K_D = 314 \pm 85$ nM and stoichiometry of 1 binding site per ca. $143 \pm 16$ Aβ42 monomers ($R^2 = 0.96$). Sample sizes consisted of 2–3 repeats per condition. **d** Kinetic aggregation profiles of 3 μM Aβ42 with 1% seed and increasing total concentrations $C$ of Brichos. Lines indicate global fits using a rate law for Aβ42 with an effective secondary nucleation rate constant $k_2(C)$ (see Methods "Kinetic modelling"; adjusted MSE = 0.00052). The sample size consisted of 1 repeat per condition. **e** Decrease in $k_2(C)$ with increasing $C$ (red data, left axis) and fractional fibril coverage by Brichos (blue data, right axis). Superimposed are model projections using parameters determined in **c** and **d** selective binding to secondary nucleation sites, $k_2(C)/k_2(0) = 1/(1 + C/K_D)$ (red line; $R^2 = 0.99$), and Langmuir isotherm (blue line; $R^2 = 0.97$). Both closely match the data. Conditions: 20 mM sodium phosphate, 0.2 mM EDTA, pH 8.0 and 21 °C.

therefore obtain sub-stoichiometric inhibition of secondary nucleation by Brichos, with almost complete suppression of secondary nucleation obtained with only one fibril-bound chaperone molecule per about 150 monomers. These results imply that Brichos binding sites and secondary nucleation sites are one and the same. Under these fibril formation conditions, secondary nucleation sites therefore have a stoichiometry of $s = 1/143$, i.e. a frequency of approximately 1 site per 150 monomers in the fibril. With four monomers per plane with two monomers per filament plane and two such filaments in an Aβ42 fibril[44,45], this stoichiometry corresponds to one catalytic site per 36 fibril planes on average. A recent evaluation of the density of fibril protrusion during an ongoing Aβ42 aggregation process at pH 6.8 found a distribution peaking at around 1 protrusion per 28 planes, or per 112 monomers, i.e. similar to the rarity of secondary nucleation sites found here at pH 8.0[46].

**Fibrils possessing fewer growth defects can be generated without measurably altering morphology**
The low stoichiometry of Aβ42 secondary nucleation sites we have found in this study supports the hypothesis that these sites are rare growth defects in the fibril core structure. To verify this hypothesis, we sought to produce Aβ fibrils under conditions designed to reduce defect stoichiometry by removing their kinetic trapping, so we could subsequently test whether their secondary nucleation sites are also reduced.

Removal of supersaturation alone, e.g., by allowing an aggregation reaction to proceed to completion, is not expected to remove kinetic trapping of defects. This is because the prevalence of long fibrils (often containing >10⁵ monomers) leads to very slow exchange of protein in the interiors of fibrils with the residual

monomeric protein in solution[28–30,47,48]. The need to depolymerise tens of thousands of monomers from the fibril to access the average defect imposes an insurmountable kinetic free-energy barrier for their removal. Therefore, fibril growth defects will persist in mature fibrils for very long periods of time unless experimental conditions are chosen to almost completely disassemble the fibrils. Also, in situ self-repair of defects is expected to be very slow even upon thermal treatment, since re-folding of defective monomers and alteration of their highly directional bonding to their neighbours is likely impractical without fibril disassembly. Instead, a more practical approach is to prevent kinetic trapping in the first place by growing the fibrils at very low supersaturation (Fig. 3a, b).

To do so, we took advantage of the fact that fibril solubility—the monomer concentration at which elongation and dissociation rates equalise—can be altered by changes in temperature. A previous study[49] reported a steady increase in Aβ40 solubility with temperature from 37 to 64 °C in 20 mM sodium phosphate, pH 7.4 (Fig. 3c, Supplementary Table 2). We validated this trend by measuring the solubility of Aβ40 at 27, 45, 50, and 60 °C under the same buffer conditions using NMR. In combination with the published solubility value at 37 °C[50], these measurements confirmed that Aβ40 solubility increases steadily with temperature from 37 to 60 °C (Supplementary Table 1). An annealing-type procedure involving incubating fibril seeds with a monomeric solution at a concentration slightly above the 60 °C solubility, and gradually reducing the temperature from 60 to 37 °C should then enable fibril growth at almost zero supersaturation (Fig. 3c). Such annealed fibrils are expected to have far fewer growth defects than fibrils produced from a similar (and therefore highly supersaturated) initial monomer concentration at

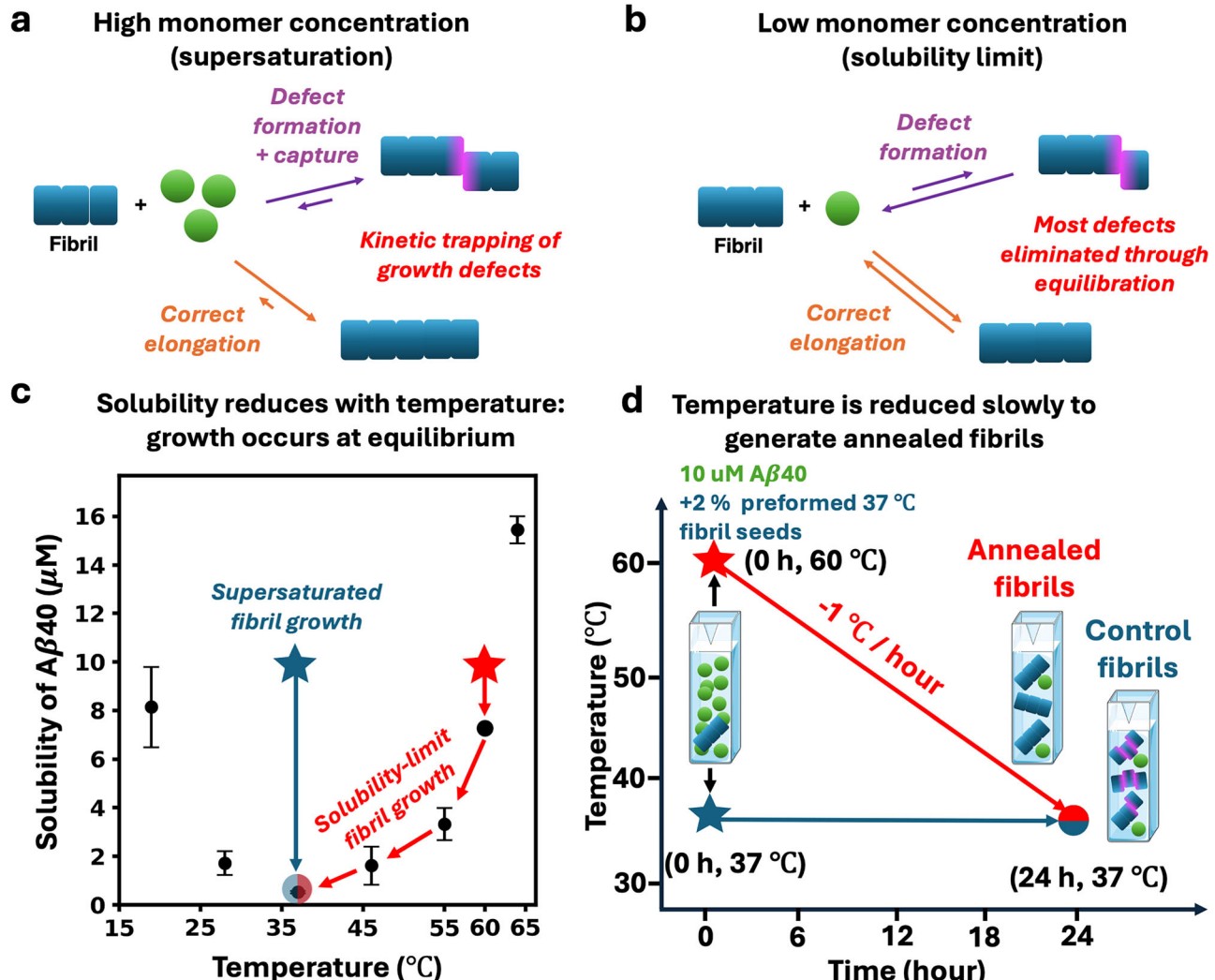

**Fig. 3 | Controlling fibril growth defect frequency by varying supersaturation levels. a** Defects (purple) are initially formed by incorrect or incomplete binding of an incoming monomer onto the fibril end and subsequently kinetically trapped by attachment of further monomer layers on top of the misaligned monomer. Defects have been illustrated as lateral dislocations of the entire cross-section for convenience only: other defect structures are also possible. **b** at low enough monomer concentrations the rate of correct elongation and dissociation are close to equilibrium (the solubility limit). In this regime defect removal is faster than formation, and kinetic trapping is largely eliminated. **c** Aβ40 solubility decreases substantially with temperature from 64 to 37 °C ensuring elongation remains close to equilibrium at all times. Data are adapted from ref. 49, except for the solubility at 60 °C, which we measured by NMR as shown in Supplementary Fig. 3. **d** We grow fibrils designed to be largely free of kinetically trapped growth defects, by slowly reducing the temperature of a solution of initially slightly supersaturated monomeric Aβ40 from 60 to 37 °C by 1 °C per hour. This ensures production of significant quantities of fibrils without requiring supersaturation.

37 °C. Here, we worked with Aβ40 rather than Aβ42 due to its higher solubility, making it easier to generate sufficient annealed fibril concentrations at low supersaturation.

First-generation Aβ40 fibrils were prepared by full aggregation of 57 μM Aβ40 monomer at 37 °C. Second-generation annealed and control fibrils were produced by incubating 0.2 μM monomer-equivalent of the first generation fibrils with 10 μM fresh Aβ40 monomer. To make annealed fibrils, the incubation temperature was started at 60 °C and was reduced by 1 °C every hour until reaching 37 °C (Fig. 3c and Supplementary Fig. 4c). To make control fibrils, the incubation temperature was maintained at 37 °C throughout. Seeding bypasses primary nucleation[43,51], and at the low supersaturation used to make annealed fibrils, elongation dominates over secondary nucleation. Therefore, most annealed fibrils are expected to be elongated first-generation seeds. Since elongation typically preserves fibril morphology[52], using first-generation seeds therefore greatly reduces the risk that annealed and control fibrils will have different morphologies.

To confirm that the annealing procedure does not cause morphological changes, we performed high-resolution cryo-electron microscopy on both annealed and control fibrils. By design, the high-resolution algorithms can only detect regular, periodic structural features of amyloid fibrils. No matter how large a sample is analysed, irregular, nonperiodic defects can only ever appear as noise to the algorithms, and are therefore averaged away during image processing[53]. We observed no significant differences in the raw images (Fig. 4a, b), and collated 2D class averages were indistinguishable between conditions (Fig. 4c-f). We finally collected statistics on these classes, and found that the proportions of twisted vs straight fibrils were almost identical in each sample (43:57 in annealed fibrils and 40:60 in control fibrils). Furthermore, the average crossover length was >300 nm in both samples, and the fibril widths matched within error. In other words, there are no detectable morphological differences between annealed and control fibrils. The only possible major differences between the annealed and control fibrils are therefore fibril length, and defect frequency and structure.

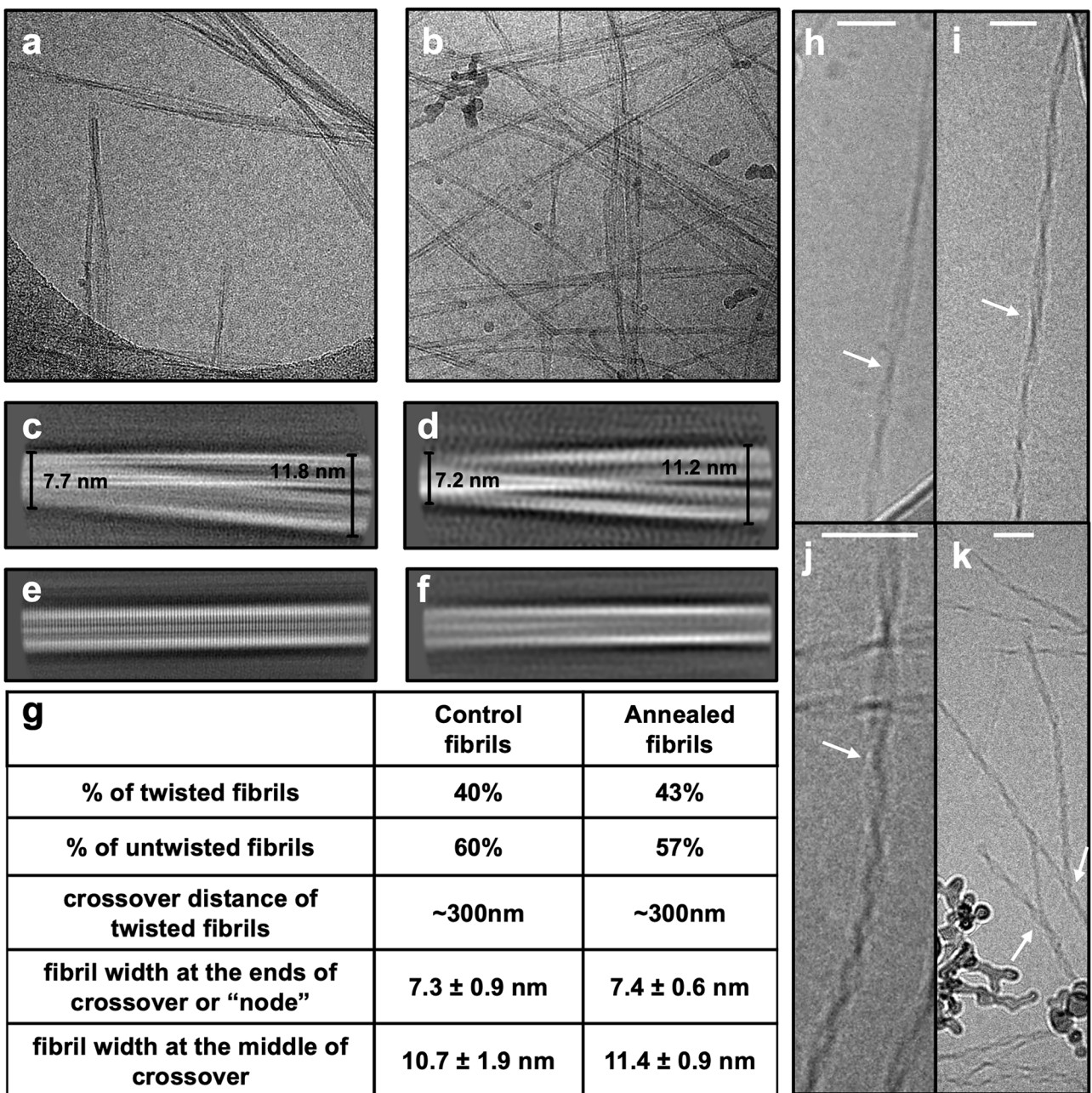

**Fig. 4 | Cryo-EM data processing and statistics of control fibrils vs annealed fibrils. a, b** Motion corrected micrographs showing control fibrils and annealed fibrils, respectively. **c, d** 2D class averages for twisted fibrils in control fibrils and annealed fibrils datasets, respectively, and measurements of fibril diameter at points along the fibril crossover. **e, f** 2D class averages for untwisted fibrils in 37 °C fibrils and annealed fibrils datasets, respectively. **g** summary of fibril morph population and morphology statistics. Cryo-TEM images of Aβ40 (**h**) or Aβ42 (**i, j** and **k**) fibrils showing evidence of growth defects, as pointed by the white arrows. The scale bar shown as white line is 50 nm. The Aβ40 fibrils in (**h**) are formed in a solution of 10 μM Aβ40 at pH 7.4, 37 °C, while Aβ42 fibrils in (**i, j** and **k**) are formed in a solution of 5 μM Aβ42 at pH 8, 37 °C. White arrows point to the locations of the defects. Further details of the cryo-EM processing workflow are found in Supplementary Fig. 7.

Despite their invisibility to the high-resolution cryo-EM algorithms, the existence of growth defects in Aβ40 and Aβ42 fibrils can nonetheless be confirmed directly from the raw cryo-EM images. This is because certain types of growth defects produce nonlocal signatures visible in raw micrographs. These include defects that punctuate transitions from one fibril morphology to another possessing a different twist length (Fig. 4h-j), and branching sites (Fig. 4k). However, their quantification using the raw images is impractical due to their rarity combined with most fibrils being located in large, un-analysable clumps and tangles on the TEM grid[51]. More importantly, this count would miss all growth defects that affect only the adjacent fibril planes,

such as dislocations and partially misfolded monomers, and that could conceivably greatly outnumber visible defects.

## Annealed fibrils possess far fewer secondary nucleation sites than normal fibrils

Since Brichos also strongly inhibits Aβ40 aggregation at very low stoichiometries[37], it is likely that, as was shown for Aβ42 fibrils, Brichos binds almost exclusively to secondary nucleation sites and not elsewhere along Aβ40 fibrils. We therefore again sought to quantify the number of secondary nucleation sites for both annealed and control fibrils by measuring the Brichos binding stoichiometry to Aβ40 fibrils.

This time, we used fluorescence correlation spectroscopy (FCS) in addition to MDS to eliminate the possibility that differing Brichos binding stoichiometries for annealed and control fibrils are an artefact of either experimental technique.

In both FCS and MDS experiments, we mixed different concentrations of Alexa488-Brichos with fixed concentrations of non-fluorescent Aβ40 fibrils and subsequently measured the free Alexa488-Brichos concentration. In FCS this was done by positioning the confocal volume at a height where only diffusible Alexa488-Brichos can be detected, excluding those attached to the sedimented fibrils at the bottom of the dish (see Methods "Fluorescence correlation spectroscopy"). In MDS, this was done by centrifuging the samples to remove fibril-bound Brichos and performing measurements on the supernatant (see Methods "Microfluidic Diffusional Sizing"). Visual inspection of the resultant binding curves of free vs total Brichos (Fig. 5a, b) reveals annealed seeds to have a lower Brichos saturation concentration compared to control seeds, indicating a lower secondary nucleation site frequency, and supporting the hypothesis that they are growth defects.

We then globally fit these two binding curves using a simple protein-ligand binding model[54](Eq. 5). We found one Brichos binding site per 99 monomers in control fibrils. This, alongside the fact that an additional and distinct site quantification technique was used compared to Aβ42, provides even more support for the rarity of Aβ secondary nucleation sites. In annealed fibrils, the stoichiometry of these sites was reduced to one per 834 monomers. This implies that the number of secondary nucleation sites is reduced by approximately 88% by the annealing treatment relative to fibrils formed entirely at 37 °C. This large reduction validates a key prediction of the hypothesis that the secondary nucleation sites are growth defects.

Note that, if Brichos also binds elsewhere along Aβ40 fibrils, the secondary nucleation site stoichiometry would be even lower than what was inferred here from the Brichos binding stoichiometry. Since the annealed and control fibril morphologies have been demonstrated to be the same on average, this would in turn imply an even greater reduction in secondary nucleation site incidence in annealed fibrils compared to control fibrils if they are indeed growth defects. Trimerization of Brichos in solution[38] could scale the measured free Brichos concentrations by a constant factor, but this will not affect the fitted binding stoichiometry, instead only changing the y-axis proportionality constants in our binding model.

### Annealed fibrils display considerably reduced secondary nucleation seeding efficiency compared to normal fibrils

We next used 0.1 or 2 μM monomer-equivalent of the second generation fibrils (control seeds or annealed seeds) as preformed seeds in new aggregation reactions of 5 μM monomeric Aβ40 at 37 °C. We find that even though both control and annealed seeds accelerate the aggregation (Supplementary Fig. 1), the annealed seeds have much less seeding efficiency compared to control seeds in both lightly and heavily seeded conditions, Fig. 5c, d.

The decrease in seeding efficiency of annealed seeds compared to control seeds could, in principle, reflect annealed fibrils having greater length, and thus possessing fewer elongation-competent fibril ends per unit fibril mass, although such length differences should be limited by the iterative seeding approach used in their production. To test this possibility, aggregation data were fitted using the standard kinetic model for Aβ40 aggregation[16], allowing the seed fibril length to differ between control seeds and annealed seeds (by allowing the seed fibril number concentration $P_0$ to differ between conditions at fixed seed mass concentration $M_0$). This model was unable to globally fit the data, even assuming extreme (≫trillion-fold) length differences, yielding pronounced misfits particularly at 2 μM monomer-equivalent of seeds (Fig. 5c). In fact, lower-resolution cryo-TEM images show no sign that annealed and control fibrils differ significantly in length

(Supplementary Fig. 5). More realistic upper bounds of their average length difference (e.g. 10× or 100×) lead to markedly worse misfits (Supplementary Fig. 5), further strengthening the conclusion that length differences alone cannot explain the large difference in seeding propensity of these fibrils. Note that directly eliminating seed length differences by sonication was not feasible, because Aβ40 fibrils generated under standard in vitro conditions are highly resistant to sonication[51]. Moreover, vigorous sonication of annealed and control fibrils in weaker fibril types such as Aβ42[51] may never reduce them to the same length if fragmentation occurs preferentially at defect sites (see Supplementary Discussion).

The reduced number of secondary nucleation sites in annealed fibrils can be captured by a kinetic model that is identical in all respects to the standard model for Aβ40 aggregation[16], but where the seed mass concentration $M_0$ is multiplied by fitting parameter $r$, the fraction of remaining secondary nucleation sites compared to control fibrils (See Methods Sec. "Kinetic modelling" for modified kinetic model equations). We find that such a model, where effectively $M_0$ can vary but not $P_0$, can fit the measured kinetic data very well (Fig. 5d), using a value for $r$ of 0.08, which is consistent with a 92% reduction in secondary nucleation site numbers. This is remarkably consistent with the 88% reduction estimated using Brichos stoichiometry and is despite this model having no more degrees of freedom than the differing-seed-length model. In general, one might expect the ensemble of defect structures in annealed and control fibrils to differ somewhat. However, this close correspondence between secondary nucleation site stoichiometry and secondary nucleation propensity implies that there is little structural difference on average between the secondary nucleation sites in annealed and control Aβ40 fibrils.

### Aβ40 secondary nucleation occurs predominantly at defects regardless of the supersaturation level

Brichos can increase the Aβ40 aggregation half-time by at least 10-fold under the conditions used to generate the control fibrils[37], which, given Aβ aggregation half times scale inversely with the cube root of the secondary nucleation rate[55,56], implies a ≥1000-fold reduction in total secondary nucleation. As demonstrated above, Brichos effects this inhibition by blocking defect sites, so its causing this ≥1000-fold reduction requires that nucleation from non-defect surface sites on control fibrils must be at least 1000-fold slower than nucleation at defects in the absence of Brichos. Annealed Aβ40 fibrils grown at near-zero supersaturation still possess growth defects at Boltzmann distribution-governed stoichiometry, and their secondary nucleation rates and defect abundance are reduced by 8- to 12-fold relative to control fibrils. Consequently, even in supersaturation-free Aβ40 fibrils, secondary nucleation at defects remains at least about 50-fold faster than nucleation from other, non-defect surface sites under the same conditions.

At equilibrium, the fraction of defective fibril planes vs total fibril planes $p_{eq}$ is mathematically related to the free energy penalty $\Delta G_{def}$ for forming the least unstable possible types of defect by $-RT\ln(p_{eq}) = \Delta G_{def}$ (assuming defective layers are rare, see Supplementary Methods Sec. 3.1). $p_{eq}$ is related to the equilibrium defect site stoichiometry per monomer $s_{eq}$ by $s_{eq} = n_{def}p_{eq}/x$, where x is the number of monomers in the fibril cross-section and $n_{def}$ the number of discrete defect sites per defective layer (Supplementary Methods Sec. 3.1). For instance, $n_{def} = 2$ in dislocation-type defects because dislocations expose fibril core on both sides of the fibril. In Supplementary Methods Sec. 3.1, we use these numbers to estimate the equilibrium defect-free energy penalty for several plausible values of $x$ and $n_{def}$, finding $\Delta G_{def}$ to range from −14.1 to −17.8 kJ/mol. This being many multiples of the thermal fluctuation energy $RT$, these defects are thus very thermodynamically unstable, as expected given their low stoichiometry. $\Delta G_{def}$ is also proportionally significant: approximately half of the per-monomer free energy for correct elongation (calculated

**a** FCS confirms annealing reduces 2° nucleation sites

**b** MDS confirms annealing reduces 2° nucleation sites

**c** Kinetic model: Different seed lengths only

**d** Kinetic model: Most 2° nucleation sites removed

Fig. 5 | **Annealed fibrils have fewer secondary nucleation sites and lower secondary nucleation rate than control fibrils. a** and **b** Solutions of 4.5 μM (**a**) or 5 μM (**b**) Aβ40 fibrils (annealed or control) with 3 to 100 nM Alexa488-Brichos were monitored by FCS (**a**) or MDS (**b**) to see the number of diffusing Alexa488-Brichos. Global fitting of both FCS and MDS data with a standard ligand-protein 1:1 binding model (Eq. (5)) reveals the number of Alexa488-Brichos binding sites per Aβ40 molecule in the fibril for annealed seeds and for control seeds is $s=1/99$ (95% Confidence Interval = [1/116, 1/89]) and $s=1/834$ (95% Confidence Interval = [1/1216, 1/603]), respectively. Solid lines show the bootstrap median fit. The shaded regions represent bootstrap confidence bands, which illustrate the uncertainty in the model predictions across resampled datasets, as detailed in Methods Sec. "Fitting of ligand-protein binding model". Sample sizes consisted of 1–3 repeats per

condition. **c**, **d** The aggregation of 5 μM Aβ with 0.1 or 2 μM monomer-equivalent of annealed or control seeds was monitored by ThT fluorescence in a plate reader and globally fitted to kinetic models (lines). The model in (**c**) allowed annealed seeds to differ from control seeds only by their length and gave misfits with even a 100 trillion-fold average length difference. Using more plausible average fibril seed length differences (10–1000x) leads to even worse misfits (Supplementary Fig. 5). Sample sizes consisted of 2–3 repeats per condition. The model in **d** allowed annealed seeds to differ from control seeds only by their secondary nucleation site stoichiometry (see Methods Sec. "Kinetic modelling") and gave good fits. Again, defects have been illustrated schematically as lateral dislocations for convenience only.

in Supplementary Methods Sec. 3.1 as − 32.3 kJ/mol at 45 °C). Note this calculation applies to the majority of defect structures seen in annealed fibrils. However, as shown above, there is evidence to suggest that the defect structures in annealed and control fibrils are similar.

If $\Delta G_{def}$ for Aβ42 fibrils is around half of the per-monomer free energy for correct elongation, as it is for Aβ40 fibrils, then switching to non-supersaturated growth conditions can be calculated to again reduce growth defect frequency by around 20-fold. Nonetheless, Brichos binding can also increase Aβ42 aggregation half-times by at least 6.5-fold[46], corresponding to a >250-fold reduction in secondary nucleation rate. So, fibril growth defects should remain the dominant source of both Aβ40 and likely also Aβ42 secondary nucleation even without any supersaturation. (Note, older and less detailed kinetic models predict that the aggregation half times scale inversely with the

square root of the secondary nucleation rate instead of the cube root[10,16]. Even if this were true, however, our conclusion would remain unchanged.)

## Discussion

There are several reasons to expect that defects may act as secondary nucleation sites across many or even most types of amyloid fibrils. Diverse amyloid proteins form fibrils with highly varied surface chemistries. About the only structural feature these fibrils have in common is their hydrogen bonding between adjacent planes in the filaments, with significant differences in the packing of hydrophobic side-chains in the fibril core. The almost universal occurrence of rapid secondary nucleation across amyloid-forming proteins[9,57] thus already suggests that secondary nucleation may have more to do with the

amyloid core than with the fibril surface. Indeed, it has not yet been found to be possible to design an Aβ42 mutant that eliminates secondary nucleation without also eliminating fibril growth[11]. Moreover, substitutions outside the amyloid fibril core do not strongly affect the ability of mutant fibrils to catalyse secondary nucleation of WT Aβ42[58,59]. Another structural feature shared by all amyloid fibrils is the presence of growth defects, which is guaranteed by the laws of thermodynamics (the Boltzmann distribution) regardless of their exact structure. Since these defects can provide partial access to the fibril cross-section, they are a very strong candidate for secondary nucleation sites. Such defects can provide a partial scaffold to surmount the entropic penalty associated with forming a new oligomer or fibril[60–62]. Additional support for this proposal is found beyond the field of amyloid fibril self-assembly: analogous defect-mediated secondary nucleation is well documented in metal and polymer crystallisation[63–65].

Brichos potently inhibits secondary nucleation by binding to fibril surfaces across multiple amyloid systems, including α-synuclein[66], IAPP[67], Aβ40, and Aβ42. In both α-synuclein and IAPP, inhibition causes a greater than tenfold increase in aggregation half-time at a Brichos:monomer ratio of around 1:10. These measurements imply that only a small fraction of fibril surface area—likely rare structural defects—drives secondary nucleation under these conditions, as established above for Aβ fibrils. The ability of Brichos to bind tightly and block such sites across diverse amyloids suggests that these nucleation-prone regions share structural features. Since amyloid surface sequences and properties vary widely, these shared features plausibly involve partial exposure of the cross-β core. If so, and if growth defects universally promote secondary nucleation, then Brichos may be an even more general secondary nucleation inhibitor than is currently appreciated. Possession of a universal affinity for cross-β structure would also explain why Brichos often displays at least some ability to inhibit elongation by binding fibril ends[66,68], which share structural similarities with defect sites.

A surface need not share many structural features with aggregates to promote their nucleation. Indeed, primary nucleation of fibrils (and unrelated aggregates) almost always happens on reaction vessel surfaces in vitro, not in bulk[57,69,70]. By extension, other areas of the fibril surface may also act as secondary nucleation sites in some systems, in addition to growth defects. Regardless of whether significant off-defect nucleation occurs, multiple binding sites with diverse affinities likely exist on typical fibril surfaces. For example, binding studies on both Aβ42 fibrils and α-synuclein fibrils at physiological temperature and pH have revealed two classes of Brichos interaction: a rare, high-affinity site with nanomolar $K_D$, and a much weaker micromolar-$K_D$ site that may represent nonspecific surface binding[24,66,71]. In both systems, however, the high potency of secondary nucleation inhibition by Brichos[21,66] implies that secondary nucleation occurs only at the tight site (see Supplementary Table 3 for $K_D$ comparison table). These results suggest a conceptual division between defect-associated nucleation (tight sites) and more generic surface adsorption (loose sites).

Conversely, off-defect secondary nucleation along the whole fibril has been proposed for α-synuclein under acidic conditions[38]. While substoichiometric inhibition by Brichos in that case was proposed to be caused by a large diffusion-driven effective volume for Brichos reliant on relatively weak Brichos binding affinity, the experimental observations are also consistent with a defect-based pathway. Importantly, binding studies that assume a single class of sites can yield apparent average affinities or fail to detect rare high-affinity sites altogether. Kinetic modelling, therefore, provides a crucial complementary approach to determine inhibitor affinities at nucleation-active sites.

The relationship between fibril morphology and growth defects merits further study. The fibril core structure exposed at a defect could potentially be templated onto the newly forming fibril during secondary nucleation, offering a possible molecular pathway for strain propagation. A recent investigation of Brichos binding to Aβ42 fibrils[24] found that full inhibition of secondary nucleation by Brichos gives rise to a different, thinner morphology with a single filament consisting of two monomers per plane, as opposed to the more usual two-filament morph. This could indicate that secondary nucleation of Aβ42 propagates structure in some way. Alternatively, it could point to a key role of growth defects such as dislocations in anchoring the higher-order assembly of filaments. After secondary nucleation, the most notable secondary processes that can accelerate protein aggregation are fibril branching and fibril fragmentation. It is plausible that fragmentation is made possible by growth defects, which provide mechanically weak loci in fibrils. If so, and considering that rare fibril branching is also a defect-enabled phenomenon, the formation of growth defects could be the key molecular event driving autocatalytic amplification in all types of fibril formation reactions. In the Supplementary Discussion we suggest potential future experiments for testing this hypothesised relationship between defects and fragmentation, and explain why fragmentation cannot be a significant sink for defects without vigorous sonication.

Crucially, the finding that secondary nucleation sites are defects could make structure-based drug design approaches possible for developing Alzheimer's and Parkinson's disease therapeutics. Secondary nucleation sites on fibrils have long been recognised as a promising target for rational design of therapeutics[21], being both the major source of toxic amyloid-β oligomers[55], and obligate for rapid fibril proliferation[9]. Indeed, much of the interest in Brichos itself stems from its ability to bind to and block these sites to reduce the rate of secondary nucleation[38]. It also implies that reducing supersaturation levels in the brain (by e.g. reducing amyloidogenic protein production) could be a particularly effective alternative therapeutic approach that not only directly lowers aggregation rates but also reduces defect stoichiometry, leading in turn to the formation of fewer toxic oligomers. Aβ40 solubility in vivo is expected to differ substantially from in vitro measurements because chemical compositions there are very different to standard in vitro buffers. Clearly, physiological concentrations of Aβ40 must be above the local solubility limit for Aβ40 fibrils to accumulate in the brain. Since the central conclusion—that secondary nucleation predominantly occurs at growth defects—does not rely on supersaturation, it remains compatible with physiological settings even where local concentrations approach the solubility.

In summary, our study shows that amyloid-β fibrils contain a small number of secondary nucleation sites, which can be further reduced by decreasing the level of supersaturation. These nucleation sites are identified as defects formed during fibril growth. The free energy penalty for a growth defect is roughly 50% of the per-monomer elongation free energy. A deeper understanding of defect-driven secondary nucleation could open new opportunities for targeting amyloid formation in various diseases. By inhibiting secondary nucleation or reducing fibril defects, it may be possible to lower the formation of toxic species and slow disease progression. These findings offer a broader perspective on the molecular mechanisms governing amyloid-related diseases and offer a potential starting point for structure-based drug design for inhibiting amyloid secondary nucleation.

## Method

Independent repeats in the plots correspond to separate measurement acquisitions performed on the same prepared sample, rather than measurements on independently prepared samples.

### Chemicals and consumables

All Aβ42 or Aβ40 related experiments were carried out in 20 mM sodium phosphate buffer (sodium dihydrogen phosphate dihydrate and di-sodium hydrogen phosphate) at pH 8.0 or pH 7.4, respectively.

The buffer components were purchased from Sigma-Aldrich (St. Louis, MO, USA), and all buffer solutions were prepared using Milli-Q water. The buffer was filtered through wwPTFE 0.2 μm 50 mm disc filters (Fisher Scientific, Pittsburgh, PA, USA) and degassed prior to use. The dye Thioflavin T (ThT) for monitoring amyloid formation was obtained from Calbiochem. Axygen® MaxyClear Snaplock Microtubes and Corning® 96-well Half Area Black/Clear Flat Bottom PEGylated Polystyrene Microplates (3881) were used as containers for protein solutions to minimise protein adsorption onto container surfaces.

### Expression and purification of Aβ42, Aβ40 and Brichos

Recombinant peptide Aβ (M1-42) (MDAEFRHDSGYEVHHQKLVF-FAEDVGSNKGAIIGLMVGGVVIA), referred to here as Aβ42, Aβ(1-40) (DAEFRHDSGYEVHHQKLVFFAEDVGSNKGAIIGLMVGGVV), referred to here as Aβ40, and Aβ(M1-40) (MDAEFRHDSGYEVHHQKLVF-FAEDVGSNKGAIIGLMVGGVV) were expressed as inclusion bodies in *E. coli* BL21(DE3)pLysS Star and purified using denaturation, anion-exchange chromatography, and size-exclusion chromatography to obtain highly pure monomeric peptide[72,73]. Aβ(1-40) was expressed in fusion with the EDDIE tag to enable a free Asp1, and its purification required additional steps for refolding/autocleavage of the EDDIE tag[73]. The Aβ(1-40) peptide was used in all Aβ40 experiments, except in the measurements of solubility using NMR spectroscopy, in which case Aβ(M1-40) was used. The proSP-C BRICHOS domain was expressed in *E. coli* BL21(DE3), isolated under denaturing conditions by Ni$^{2+}$ affinity chromatography, and further purified following thrombin cleavage and ion-exchange chromatography[37]. Purified Aβ peptides were aliquoted and stored at −80 °C. Purified BRICHOS was aliquoted and stored at −80 °C until use. Protein concentrations were determined by UV absorbance.

Immediately prior to each experiment, Aβ42 or Aβ40 powder was dissolved in 6 M guanidine hydrochloride (pH 8.5), and monomers were isolated by size exclusion chromatography (Superdex 75 column) in 20 mM sodium phosphate buffer as described in chemicals and consumables section.

ProSP-C Brichos domain was covalently labelled with the Alexa-488 dye (Thermo Fisher Scientific, Waltham, US) by incubating the protein for three hours in 20 mM sodium phosphate buffer, pH 8.0, at room temperature under gentle agitation with one equivalent of Alexa-488 carboxylic acid succinimidyl ester to label covalently primary amines. Following the labelling reaction, excess dye was removed by desalting on a G25 gel filtration column, followed by purification by anion exchange to remove any protein with multiple labels, and an additional desalting step on a G25 gel filtration column before use. The labelling efficiency of Alexa-488 on Brichos is around 90% as indicated by relative absorbance measurements at 280 and 490 nm, see Supplementary Fig. 6. Stated Brichos concentrations always refer to both labelled and unlabelled chaperone monomers together.

### Microfluidic diffusional sizing

The fabrication and operation of the microfluidic diffusion device followed established procedures[39,40]. Briefly, the microfluidic chips were fabricated by using standard soft lithography techniques into polydimethylsiloxane[74]. The analyte sample and the buffer were introduced in the device through reservoirs connected to the inlets. The flow rates in the channels were controlled by applying a negative pressure at the single outlet channel by means of a syringe pump (neMESYS, Cetoni GmbH, Korbussen, Germany). Typical flow rates were in the range 60 μl/h to 90 μl/h and lateral diffusion profiles were recorded at twelve different positions (3.52, 5.29, 8.57, 10.33, 18.61, 20.37, 28.65, 30.41, 58.69, 60.45, 88.73 and 90.5 mm) by standard epifluorescence microscopy using a cooled CCD camera (Evolve 512, Photometrics, Tucson, AZ, USA). The diffusion profiles were fitted by model simulations based on advection-diffusion equations under the assumption that the diffusivities of the species in the mixture are distributed according to a bimodal Gaussian distribution[39]. In this approach, the fitting parameters are the mean and the standard deviation of at least two size distributions, corresponding to the populations of the free chaperone (small size range) and of the chaperone bound to the fibrils (large size range).

In order to evaluate the binding between Brichos and Aβ42 fibrils during the aggregation process, samples with Alexa-488 labelled Brichos at concentrations of 0.25, 0.35 or 0.50 μM were mixed with 24 μM Aβ42 monomers and 20 μM ThT and incubated in a 96-well plate in a Fluostar Optima plate reader (BMG Labtech, Ortenberg, Germany) at 37 °C with double orbital rotation (400 rpm) in order to generate sufficiently short fibrils for injecting into the microfluidic channels[10]. The aggregation kinetics were monitored by following the associated increase in ThT fluorescence. After completion of the reaction, the samples were incubated for at least 2 days at 21 °C to ensure equilibrium conditions. They were subsequently analysed by microfluidic diffusion sizing at 21 °C.

To evaluate the binding between Brichos and mature Aβ42 fibrils, the same Aβ42 aggregation reaction was performed at 37 °C but without Brichos. After completion of the reaction, aliquots of Alexa-488 labelled Brichos in the concentration range between 0 and 1.5 μM were added to samples of the mature Aβ42 fibrils and incubated for at least 2 days at 21 °C to ensure equilibrium conditions. They were subsequently analysed by microfluidic diffusion sizing at 21 °C. To better constrain the Langmuir model during subsequent fitting, more measurements were taken at lower total Brichos concentrations, where the bound concentration changes more rapidly.

Brichos binding to Aβ40 fibrils was evaluated by incubating 5 μM Aβ40 fibrils (control or annealed) with Alexa-488 labelled Brichos at 12.5, 25, 50 or 100 nM for 10 min at 37 °C. Before loading the samples into MDS, Aβ40 samples were centrifuged and only supernatant was injected, as Aβ40 fibrils are shake-resistant and remain too long in the microfluidic channels[51]. Samples were centrifuged at 17,850 rpm (~29,000 $g$) for 15 min at 37 °C, and 4 μL of each supernatant was loaded into microfluidic channels for analysis on a Fluidity One-M Serum instrument (Fluidics Inc., Cambridge, UK). The intensity of Alexa-488 was measured, and the artefactual contribution from free dye was removed using the free dye fractions determined in the FCS experiments.

### Kinetic experiments

The kinetic assay for Aβ40 displayed in Fig. 5b, c starts with 5 μM monomer in 20 mM pH 7.4 phosphate buffer with 10 μM ThT, with or without 0.1 or 2 μM monomer-equivalent of seeds. The seeds were generated as shown in Supplementary Fig. 4. After mixing the monomer and the seeds, the solutions were immediately incubated in a 96-well half-area clear flat-bottom PEGylated polystyrene microplate at 37 °C. The fluorescence was recorded in a Fluostar Omega plate reader (BMG Labtech) using a 440 nm excitation filter and a 480 nm emission filter, and used to monitor the extent of aggregation. The waiting time between the reading cycles was set to 0, meaning that the wells were continuously monitored throughout the kinetic measurements. This setup is designed to control the agitation frequency caused by reading wells consistently across all experiments[46].

The kinetic assays for Aβ42 were performed by incubating a solution of 3 μM Aβ42 peptide, 1% seeds, and 20 μM ThT with or without different concentrations of Brichos in a non-binding 96 well plate in a plate reader Fluostar Optima (BMG Labtech, Ortenberg, Germany) at 37 °C with double orbital rotation (400 rpm). The validity of ThT fluorescence as a quantitative reporter of Aβ40 and Aβ42 fibril formation using this protocol has been established in multiple mechanistic studies[10,16,75]. Its use here complies with community requirements on chemical probes.

## Kinetic modelling

**Aβ42 aggregation.** We describe Aβ42 aggregation using a chemical kinetics framework that accounts for the various microscopic steps of aggregation (Fig. 1a)[16]. The unperturbed kinetic equations for the aggregate number concentration $P(t)$ and aggregate mass concentration $M(t)$ are:

$$\frac{dP}{dt} = k_1 m^{n_1} + k_2 m^{n_2} M, \tag{1a}$$

$$\frac{dM}{dt} = 2k_+ m P = -\frac{dm}{dt}, \tag{1b}$$

where $m(t)$ is the monomer concentration, $k_1$, $k_2$, $k_+$ are the rate constants for primary nucleation, secondary nucleation and elongation, and $n_1$, $n_2$ are the reaction orders of primary and secondary nucleation with respect to the monomer concentration.

Fits of the aggregation curves in the presence of Brichos in Fig. 2d were performed using the Amylofit platform (which implements analytical solutions to the kinetic equations (1a), (1b))[43]. The presence of Brichos was captured by means of an effective rate constant for secondary nucleation $k_2$ that depends on chaperone concentration.

**Aβ40 aggregation.** Equation (1) apply unmodified to Aβ40 aggregation seeded with control fibrils. When an Aβ40 aggregation reaction is seeded with annealed fibrils, Eqs. (1) still apply, but with modified initial condition $M(0) = rM_0$, where $M_0$ is the mass concentration of seed fibrils and $r$ is the fractional reduction in secondary nucleation sites caused by the annealing process. The analytical solution to such equations is then identical to that used in Amylofit[43], but with all instances of $M(0)$ replaced by $rM_0$. We used an offline version of Amylofit to globally fit the modified and unmodified analytical solutions to the kinetic data for unseeded reactions and for reactions seeded with annealed and control fibrils, and to determine $r$. Full fits are in Supplementary Fig. 1; Figure 5d omits the unseeded kinetic curve to provide greater resolution on the seeded kinetic curves. Misfits in Fig. 5c were performed using the standard kinetic model in the AmyloFit platform with different average seed fibril lengths allowed for the annealed and control fibrils.

## Solubility measurement

The aggregation of 106, 106, 132, and 101 μM Aβ (M1-40) monomers at 27, 45, 50, and 60 °C, respectively, was monitored by integrating the methyl group region of the Aβ (M1-40) $^1$H NMR spectra. The solubility was estimated from the relative integral of the methyl signals at the final plateau compared to the initial value. The concentration of Aβ (M1-40) at the initial value of NMR was determined using UV absorbance. A detailed description of the solubility measurements is provided in Supplementary method sec. 3.3, and the solubility of Aβ (M1-40) monomers at 27, 45, 50, and 60 °C is provided in the Supplementary table 1. The solubility of Aβ (M1-40) at 60 °C is plotted as a single dot in Fig. 3c. NMR experiments were performed on a Bruker Avance III HD 900 spectrometer (Bruker Biospin, Rheinstetten, Germany), operating at a $^1$H resonance frequency of 899.8 MHz and fitted with a 5 mm cold probe.

## Fluorescence correlation spectroscopy

Before FCS measurements, samples of 4500 nM Aβ40 fibrils and 3–100 nM Alexa488-Brichos were incubated together in MatTek dishes (35 mm, 10 mm glass bottom, No. 1.5 glass) at 37 °C for 2 h in a desiccator with water below to avoid evaporation. The measurements were conducted using a Zeiss 980 confocal laser scanning microscope (LSM 980, Carl Zeiss) using ZEN 3.10 software, equipped with a Zeiss

C-Apochromat 40×−/1.2 NA water immersion objective. Excitation was performed at 488 nm, and fluorescence emission was collected within the 499-622 nm range. Each data point in Fig. 5a is based on 3–4 FCS measurements, 90 s measurements each. FCS data were acquired using Zeiss ZEN microscopy software (Zeiss). The free Brichos concentration was determined by fitting the correlation curves at a time range of $10^{-5}$ to 1 s to theoretical models for diffusion of two species of different molecular weights, representing the diffusion of free dye and Brichos (Fig. SI2)[76], using the model:

$$Y = \frac{1}{N}\left[\frac{1-f_2}{\left(1+\frac{X}{\tau_{D1}}\right)\sqrt{1+\frac{X}{\tau_{D1}S^2}}} + \frac{f_2}{\left(1+\frac{X}{\tau_{D2}}\right)\sqrt{1+\frac{X}{\tau_{D1}S^2}}}\right] + 1$$

where $N$ is the total number of particles, $f$ is the fractional contribution of the component, $X$ is correlation time, $S$ is the structural parameter fixed to 5, $\tau_{D1}$ and $\tau_{D2}$ are the diffusion times of the two components, which are fixed to 27 μs and 126 μs, respectively, representing the diffusion time of free dye and Brichos. The resulting free Brichos particle number is shown in Supplementary Fig. 2c, and the average fraction of Brichos at the baseline or above the baseline (4% and 10%) is used to calculate the free Brichos concentration, which is fitted by the ligand-protein binding model for stoichiometry.

## Fitting of ligand-protein binding model

We consider the 1:1 binding reaction: $P + L \underset{k_{\text{off}}}{\overset{k_{on}}{\rightleftharpoons}} PL$, where our ligand $L$ is Brichos and our protein $P$ is a secondary nucleation site. At binding equilibrium, the total ligand concentration $c_L$ is:

$$c_L = [L] + [PL] = [L] + \frac{c_P[L]}{[L] + K_D}, \tag{2}$$

where $[L]$ is the concentration of free Brichos, $[PL]$ is the concentration of Brichos bound to secondary nucleation sites, $c_P$ is the total concentration of secondary nucleation sites, and $K_D$ is the dissociation constant. This can be rearranged to:

$$[L]^2 + [L](K_D + c_P - c_L) - c_L K_D = 0. \tag{3}$$

Finally:

$$[L] = -0.5 \cdot (K_D + M_0 s - c_L) + \sqrt{0.25 \cdot (K_D + M_0 s - c_L)^2 + c_L K_D}, \tag{4}$$

where $s$ is the number of secondary nucleation sites per monomer residue in the fibrils, and $M_0$ is the fibril mass concentration. To determine the number of secondary nucleation sites per monomeric residue ($s$) in Fig. 5, the free Brichos concentration $y$ as a function of total Brichos concentration $x$ was fitted with the following ligand binding equation, which is a rearranged form of Eq. (4):

$$y = a \cdot \left(-0.5B + \sqrt{0.25B^2 + x \cdot K_D}\right) \tag{5a}$$

$$B = K_D + sM_0 - x, \tag{5b}$$

where $a$ is an arbitrary proportionality constant, $K_D$ is the dissociation constant for Brichos binding, and the quantity in the brackets is the concentration of free Brichos. The concentration of Aβ40 fibrils is $M_0 = 4500$ nM in FCS experiments, and $M_0 = 5000$ nM in MDS experiments. FCS and MDS datasets were fitted globally, sharing $K_D$, $s$(annealed), and $s$(control), while allowing separate scaling factors $a_{\text{FCS}}$ and $a_{\text{MDS}}$ for the two techniques.

To assess parameter uncertainty, we used nonparametric stratified bootstrap resampling: one replicate per Brichos concentration was randomly drawn for each dataset, the global fit was repeated (Q =

1000 iterations), and the distributions of fitted parameters were summarised by medians and 95% confidence intervals. Confidence bands in the plots were generated as the spread of bootstrap-predicted curves.

## Cryo-EM sample preparation

The samples were imaged using cryo-EM as described previously[73]. Specimens were plunged using an automated plunge freezer (Leica EM GP) set at 20 °C with 90% relative humidity. Thin liquid films were created on glow-discharged lacey carbon 300 mesh grids (Agar Scientific) and blotted with filter paper before being plunged into liquid ethane at −184 °C, ensuring vitrification in a glass-like state that prevents ice crystal formation and preserves the original microstructures. The specimens were stored in liquid nitrogen until further use.

## Cryo-EM imaging

The prepared specimen grids were transferred into the electron microscope (JEM 2200FS) using a Fischione model 2550 cryo-transfer tomography holder. Imaging was performed at an acceleration voltage of 200 kV, with zero-loss images captured digitally using a TVIPS F416 camera and SerialEM 4.0.27 under low-dose conditions with a 10 eV energy filter. Image contrast was adjusted automatically using ImageJ version 1.53a.

## Cryo-EM data collection and processing

The cryoEM datasets were collected at the SciLife lab node in Umeå, Sweden, using a Titan Krios electron microscope (Thermo Fisher) operated at 300 kV with a Falcon4 detector and a Selectris energy filter (Thermo Fisher) was used operating with a 10 $e^-$V slit. Datasets of size 8106 movies and 7990 movies were collected for control and annealed fibrils, respectively, at a nominal magnification of 130,000× was set yielding a pixel size of 0.92 Å. A defocus range of −1.4 to −2.6 μm and a total dose of 40 $e^-$/Å$^2$ were used over an exposure time of 4.39 seconds.

The raw EER movies were fractionated, aligned and summed using motion correction in RELION-4[77] and CTF estimation was done for micrographs using CTFFIND4[78]. Manual picking of fibrils was done until roughly 100,000 particles of 300 pixel box size were picked, and the extracted segment was used to train a separate picking model for each dataset in Topaz, which is incorporated in RELION-4.0[77]. Iterative rounds of 2D classification were performed to remove picking artefacts, such as carbon edges and ice contamination, until only classes that could be protein fibrils remained. For 2D classification, a box size of 600 pixels was used. Twisted and straight fibrils were then separated out and a final round of 2D classification was performed separately for both sets of fibrils.

Note that, due to current software limitations, the average crossover length is too large for 3D fibril structures to be calculated; however, in this study, these 3D structures are not needed since the 2D classes already rule out significant structural differences in the fibrils.

## Reporting summary

Further information on research design is available in the Nature Portfolio Reporting Summary linked to this article.

## Data availability

Source data are provided with this paper. The raw data generated in this study have been uploaded to figshare [https://doi.org/10.6084/m9.figshare.28716485]. Source data are provided with this paper.

## Code availability

Simple Python code used to fit data in Figure 5 is available on CodeOcean [https://doi.org/10.24433/CO.3160717.v1].

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

## Acknowledgements

This work was supported by the Swedish Research Council (2019-02397 to E.S., 2015-00143 to S.L., and 2022-06641 to S.L. and E.S.), and the GenerationNano project, the European Union's Horizon 2020 research and innovation programme under the Marie Skłodowska-Curie grant agreement No 945378 (S.L. co-PI). We acknowledge support from the Wellcome Trust (T.P.J.K.), the Cambridge Centre for Misfolding Diseases (T.P.J.K.), the BBSRC (T.P.J.K.), the Frances and Augustus Newman Foundation (T.P.J.K.), the ERC PhysProt (agreement n 337969) (T.S., T.P.J.K., S.L.), ETC StG "NEPA" (A.Š. and S.C.), the Royal Society (S.C., A.S.), the ERASMUS Programme (T.S.), and The Danish Council for Independent Research | Natural Sciences (FNU-11-113326) (M.A.). This work was also funded by the Novo Nordisk Foundation (#NNF19OC0054635 to S.L.), ETH Zürich (T.C.T.M.), and the Swiss National Science Foundation (grant no 219703 to A.J.D. and T.C.T.M.). We acknowledge the use of the nano-Characterisation and nano-Manufacturing Research Equipment (nCHREM) facility for access to microscopy instrumentation. We are grateful to the late Professor Sir Christopher Dobson for invaluable conversations regarding the micro-fluidic diffusional sizing experiments. We are also grateful to Quentin A. E. Peter and Thomas Müller for their guidance on microfluidic device design. The cuvette-filled icon in Fig. 3d is by Servier [https://smart.servier.com/]. It is licensed under CC-BY 3.0 Unported [https://creativecommons.org/licenses/by/3.0/]. The authors would like to acknowledge Umeå Centre for Electron Microscopy (UCEM) for technical assistance and access to electron microscopy. Support was provided by SciLifeLab national Cryo-EM Unit at Umeå University.

## Author contributions

Conceptualisation: A.J.D., T.P.J.K., S.L., E.Sparr. Data curation: J.H., T.S., D.T. Formal analysis: A.J.D., J.H., D.T., T.C.T.M. Funding acquisition: E.Sparr, S.L., T.P.J.K., T.C.T.M., A.Š. Investigation: J.H., T.S., D.T., S.L., E.A., E.Stemme, U.L., S.W., P.A., S.C., M.A. Methodology: A.J.D., J.H., S.L., E.Sparr, T.C.T.M., T.S, J.D.S. Project administration: A.J.D., T.P.J.K., T.C.T.M., S.L., E.Sparr. Resources: S.L., E.Sparr, T.P.J.K., S.W. Software: J.H., T.S., D.T., G.M., T.C.T.M., A.J.D. Supervision: A.J.D., S.L., E.Sparr, T.P.J.K., T.C.T.M., P.A., A.Š. Validation: J.H., T.S., S.W. Visualisation: J.H., T.S., T.C.T.M., S.L., D.T. Writing - original draft: A.J.D., J.H., T.C.T.M., T.S. Writing - review & editing: J.H., T.S., D.T., E.A., E. Stemme, U.L., S.W., G.M., S.C., M.A., M.V., A.Š., J.D.S., T.P.J.K., E.Sparr, S.L., T.C.T.M., and A.J.D.

## Competing interests

The authors declare no competing interests.

## Additional information

[1]Division of Physical Chemistry, Department of Chemistry, Lund University, Lund, Sweden. [2]Department of Biochemistry and Structural Biology, Lund University, Lund, Sweden. [3]Centre for Misfolding Diseases, Yusuf Hamied Department of Chemistry, University of Cambridge, Cambridge, UK. [4]Luxembourg Centre for Systems Biomedicine (LCSB), University of Luxembourg, Esch-sur-Alzette, Luxembourg. [5]Astbury Centre for Structural Molecular Biology, School of Molecular and Cellular Biology, Faculty of Biological Sciences, University of Leeds, Leeds, UK. [6]Living Systems Institute, University of Exeter, Exeter, Devon, UK. [7]Department of Applied Physics, Biophysics Group, SciLifeLab, Royal Institute of Technology-KTH, Solna, Sweden. [8]UK Dementia Research Institute, University of Cambridge, Cambridge, UK. [9]Institute of Science and Technology Austria, Klosterneuburg, Austria. [10]Department of Biomedicine, Aarhus University, Aarhus, Denmark. [11]Department of Chemistry and Applied Biosciences, Institute for Chemical and Bioengineering, ETH Zurich, Zurich, Switzerland. [12]Department of Physics, Kansas State University, Manhattan, USA. [13]Cavendish Laboratory, University of Cambridge, Cambridge, UK. [14]Institute of Biochemistry, Department of Biology, ETH Zurich, Zurich, Switzerland. [15]Bringing Materials to Life Initiative, ETH Zurich, Zurich, Switzerland. ✉e-mail: alexander.dear@bc.biol.ethz.ch

