## [Transparent Peer Review file · Nature Communications]

Structural defects in amyloid- β fibrils drive secondary nucleation

Corresponding Author: Dr Alexander Dear

Version 1:

Reviewer comments:

Reviewer #1

(Remarks to the Author)

The manuscript from Hu et al investigates the structural origin of secondary nucleation of amyloid beta (A-beta) peptide, which is closely connected to Alzheimer's Disease. The secondary nucleation process has in several previous papers been shown to have a key role in the amyloid formation of A-beta and the manuscript thereby have important impact for understanding of the basic biochemical mechanisms behind the disease. I believe this is an important and insightful paper that should be published after addressing the following questions/issues:

1. The kinetically trapping of defects (as illustrated in figure 3) would need some further motivation. Since the binding free energy between defect monomer and the fibril is significantly lowered, it would be an obvious sites for fibril breakage (discussed on page 16, lines 513-515). This should open for preferential fibril breakage at these sites with a much faster exchange of the defect monomer or a (even faster) structural reorganisation of the monomer at the fibril end. Even in the case of a two filament fibril, partial breakage of the defect filament could allow for monomer reorganisation and re-formation of a non-defect fibril.
2. Investigating the differences in fragmentation between annealed and non-annealed fibrils could give further support for the reduced defect frequency of annealed fibrils. For example, length distributions after gentle fragmentation (stirring or gentle sonication) could reflect the frequency of defects.
3. Comparing the length distributions (statistical analysis of images from TEM or AFM) of the annealed and non-annealed fibrils (section 2.5) would also strengthen the conclusion that it is not the number of fibril ends that give rise to the observed differences in seeding capacity.
4. Much of the discussion relies on the solubility data of Ab40 presented in table S1. Why do the authors use solubility data from a 20 years old Masters thesis (most likely not peer-reviewed) when they i. have their own data from recent publications and ii. show that they can easily measure the data by 1H NMR experiments?
5. When discussing the binding stoichiometry of proSP-C BRICHOS, it should be mentioned that it under normal conditions exists as trimer and how this affect the determined binding stoichiometry (see e.g. Biverstal H, et al. *Biochim Biophys Acta* 1854, 835-843 (2015) or Willander H, et al. *Proc Natl Acad Sci U S A* 109, 2325-2329 (2012))
6. The discussion highlights the implication of the structural defects for AD pathology. What level of supersaturation is expected in vivo? Normal CSF concentrations of Abeta are clearly below the solubility limit, and the concentration decreases further during the disease. This would indicate that defects, and thereby secondary nucleation, may not have a major role in vivo.

Minor issues:

- Page 7, line 197: should it be '< 90%' ?
- Page 16, line 483: change Kd to KD
- Page 24, Ref 39: missing details
- Page 26, Ref 55: missing details
- SI, page 4, line100: should it be 'x = 2'?

SI, Figure S3: What is the temperature for the NMR data? Both 60 and 37 deg are mentioned in the legend.

(Remarks on code availability)

Reviewer #2

(Remarks to the Author)

In this manuscript the authors describe the mechanisms by which structural defects on amyloid- β fibrils act as secondary nucleation sites through analyzing binding kinetics of Brichos chaperon domain onto fibril surfaces. They also demonstrate that these defects and thereby secondary nucleation can be strongly inhibited by decreasing monomer concentration or by steadily decreasing reaction temperature. These so-called "annealed" A β fibrils are structurally indistinguishable from control fibrils and yet contain far fewer secondary nucleation sites and have much lower secondary nucleation propensity.

The manuscript is well-written and clear. Considering the recent approval of anti-A β antibodies for symptomatic treatment of Alzheimer's disease, the manuscript is quite timely and moves the field forward. No major issues in methodology or interpretation is evident, but a number of minor issues should be addressed:

1. Data shown in Fig. 2b should be further clarified. First, individual data points should be reported instead of mean \pm SD, considering that N=2 for the blue curve. Second, the accuracy of the data points for the blue curve needs to be double-checked for (i) Brichos concentrations tested for the two different conditions do not overlap, (ii) the concentrations tested for the "blue" condition are not well spaced (please pay particular attention to the 6th data point). Third, the authors should specify whether the regression line is calculated for the blue curve only.
2. Also in Fig. 2b, for the 7th Brichos concentration tested, the higher of the two data points is likely above the maximum coverage value (based on the binding stoichiometry determined from the fit curve). The authors should explain the potential causes of this apparent measurement error. Also, the coefficients of determination (R^2) should be reported for curves in Fig 2 as was done in Fig. 5.
3. In SI Section 1.2.2, for 2-monomer-thick fibril, x should be 2, not 4. Fortunately, free energy differences were calculated using x = 2; so, a textual correction will be sufficient.
4. In SI Section 3, K_D should be K_D (equations 16 and 17) to be consistent with the rest of the manuscript. The same goes for line 483 in the main text.
5. In SI Fig. 3 caption, NMR spectra is reported to be obtained at 37°C, but the text and title of the legend suggest 60°C.
6. Several paragraphs of the Results section (starting in lines 283, 295, 375 and 407) rather belong to the Discussion section, some of which contain material that are repeated in the Discussion.
7. Brichos should be adequately introduced and relevant reference(s) should be cited.
8. Fig. 1a could be cited at the end of the sentence starting in line 65.

(Remarks on code availability)

Reviewer #3

(Remarks to the Author)

This study presents a compelling investigation into the role of structural defects in amyloid- β (A β) fibrils as catalysts for secondary nucleation, a key process in amyloid aggregation associated with neurodegenerative diseases. The authors employ an impressive combination of biophysical techniques including microfluidic diffusional sizing, fluorescence correlation spectroscopy, and cryo-EM, coupled with kinetic modeling, to demonstrate that secondary nucleation sites correspond to rare growth defects in fibrils. The findings significantly advance our understanding of amyloid aggregation mechanisms and have important implications for therapeutic strategies targeting secondary nucleation.

The study's major strengths lie in its novel mechanistic insights and methodological rigor. By quantitatively showing that annealed fibrils exhibit dramatically reduced secondary nucleation activity while maintaining standard morphology, the authors provide evidence linking defects to nucleation sites.

However, I would like clarification from the authors on several important issues before recommending publication. They are listed below:

1. The demonstration that Brichos binds to these sites at substoichiometric ratios is based on accurate determination of [Sto]. However, no mention is made about the labelling efficiency of the A488-Brichos sample and there is no discussion on whether this can significantly impact the conclusion.
2. What do the Brichos concentration represent – are the reported values total protein concentrations or concentration of the labelled fractions?
3. In Fig. 2D the fibril bound Brichos % increase with concentration but data is shown only up to 1.5 M. It would be informative to see results at higher concentrations to assess saturation behavior.

4. The authors strangely use two different techniques for estimating Brichos binding to amyloid fibrils for experiments in Fig. 2 (MDS) and Fig. 5 (FCS). For proper comparison and consistency MDS and FCS should be used for both experiments.
5. The "annealing" concept, while central to the study, isn't clearly defined until the Results section and would benefit from earlier introduction.
6. While data shown to infer Brichos binding to defect sites is compelling it can benefit from a more direct experimental evidence. For example recent work using single molecule TIRF measurements have directly captured secondary nucleation events from visualisation of branching (<https://pubs.acs.org/doi/full/10.1021/jacs.1c07228>). These types of direct measurements could complement the results described here.
7. The study assumes all secondary nucleation sites as identical between annealed and control fibrils, differing only in frequency. This point needs clarification.

(Remarks on code availability)

Version 2:

Reviewer comments:

Reviewer #1

(Remarks to the Author)

The authors have adequately addressed all my comments.

(Remarks on code availability)

Reviewer #2

(Remarks to the Author)

The authors have revised the manuscript to answer to all my (minor) comments. The manuscript is well-organized and timely, and it moves the field forward. Publication is recommended.

Additional minor issues that arose during revision:

- In line 575, the centrifugation speed should be given in g-force not in rpm, since the latter depends on the instrument.
- In their rebuttal, the authors stated that they added R^2 values of the fit curves to Fig. 2 caption, but this is only true for the fit curve in panel C but not for the fit curves in panel E.
- In Fig. S5, panel B appears to have considerably better contrast than panel A. Can the authors improve the cryo-TEM image in panel A?
- Considering that the solubility data reported in Tables S1 and S2 are generally consistent with each other, I see no clear reason for replacing Fig. 3d in the original manuscript with Fig. 3c in the revised manuscript (which has less data points and is less informative overall). The manuscript would improve if the authors replaced Fig. 3c with the original Fig. 3d and added 95% CI error bars (now reported in Table S2).

(Remarks on code availability)

Reviewer #3

(Remarks to the Author)

I am happy the way authors addressed my concerns. I have no further questions.

(Remarks on code availability)

REVIEWER COMMENTS

Reviewer #1 (Remarks to the Author):

The manuscript from Hu et al investigates the structural origin of secondary nucleation of amyloid beta (A-beta) peptide, which is closely connected to Alzheimer's Disease. The secondary nucleation process has in several previous papers been shown to have a key role in the amyloid formation of A-beta and the manuscript thereby have important impact for understanding of the basic biochemical mechanisms behind the disease. I believe this is an important and insightful paper that should be published after addressing the following questions/issues:

RESPONSE: We are very grateful for the overall positive assessment of our manuscript.

1. The kinetically trapping of defects (as illustrated in figure 3) would need some further motivation. Since the binding free energy between defect monomer and the fibril is significantly lowered, it would be an obvious sites for fibril breakage (discussed on page 16, lines 513-515). This should open for preferential fibril breakage at these sites with a much faster exchange of the defect monomer or a (even faster) structural reorganisation of the monomer at the fibril end. Even in the case of a two filament fibril, partial breakage of the defect filament could allow for monomer reorganisation and re-formation of a non-defect fibril.

RESPONSE: Although it is plausible that most fibril fragmentation occurs at defects, it is not plausible that most defects cause a fragmentation event. This would cause average fibril lengths to be at most a few times higher than defect incorporation stoichiometry. Under typical production conditions, including the quiescent conditions used here, fibril average lengths are micron-scale, consisting of tens of thousands to tens of millions of monomers, but our measured secondary nucleation site stoichiometry is c. hundreds of monomers. We are also unconvinced the removal of defects by breakage of a single defective filament in a 2-fibril fibril will be significantly faster than fragmentation of the entire fibril. First, it is unclear that mechanical forces could break just one filament in the fibril. Second, the newly broken filament ends would presumably also need to detach from the neighbouring filaments for there to be sufficient space between them to enable re-folding or reattachment of the defective monomers, imposing an additional free energy barrier.

ACTION: We have added an SI section (SI Sec. 10) outlining the above arguments and signposted it in the Discussion (Page 17, Line 465-471).

2. Investigating the differences in fragmentation between annealed and non-annealed fibrils could give further support for the reduced defect frequency of annealed fibrils. For example, length distributions after gentle fragmentation (stirring or gentle sonication) could reflect the frequency of defects.

RESPONSE: Sonication or shaking can not significantly shorten average A β 40 fibril length, since A β 40 fibrils are extremely resistant to fragmentation even under heavy sonication, as shown in [50]. However, for weaker fibrils such as A β 42, this could be a viable option and we agree it is worth stating this.

ACTION: We have added material outlining these potential future experiments to the aforementioned new SI section (SI Sec. 10) and signposted it in the Discussion (Page 17, Line 469-471).

3. Comparing the length distributions (statistical analysis of images from TEM or AFM) of the annealed and non-annealed fibrils (section 2.5) would also strengthen the conclusion that it is not the number of fibril ends that give rise to the observed differences in seeding capacity.

RESPONSE: The misfits of the kinetic data in section 2.5 shows that even if the average annealed fibril length is 100 trillion times that of control fibrils, this still cannot explain the differences in seeding capacity of annealed and control fibrils. We agree with the reviewer that this argument could potentially be strengthened by using a more realistic difference in fibril length to generate even worse misfits. Statistical analysis of TEM images of Ab40 to provide a precise factor for the difference in average length is not feasible due to the tangling of the fibrils and their large length relative to the observation field even under very low magnification. However, such images can at least provide an upper bound for these differences.

ACTION: We have added a new section SI Sec.7 and figure. S5 to the SI containing low magnification cryoTEM images, showing that the annealed fibrils are not significantly shorter than the 37 degree fibrils. Also included in the supplemental figure are revised, and worse, misfits to the kinetic data assuming more realistic differences in average fibril length (10x-1000x). We reference these new findings in Results section 2.5 (Page 14, Line 332-335).

4. Much of the discussion relies on the solubility data of Ab40 presented in table S1. Why do the authors use solubility data from a 20 years old Masters thesis (most likely not peer-reviewed) when they i. have their own data from recent publications and ii. show that they can easily measure the data by ¹H NMR experiments?

RESPONSE: We originally used these data for convenience as several temperatures were studied in the same experiments. We nonetheless take the reviewer's point that this is not optimal, especially considering the peptide preparation and purification was quite different compared to our own.

ACTION: We have replaced the solubility data in the main text Fig3 with new and published solubility data from our lab. Our results (Table 1 in SI) display a similar trend to that described in the referenced Master's thesis (Table 2 in SI), showing increased solubility with rising temperature above 37 degrees. We now retain the data from the Masters thesis in the SI for comparison/validation purposes only. In the same SI section we give an expanded account of the methodology in the various experiments performed. We have also added Methods section (Solubility measurement, line 594-602, page 20), and sentences to the Results (line 225-229, page 9) explaining that we measured the solubility with NMR at temps 27, 45, 50, and 60 degrees and used our group's previously published data for 37 celsius. The new solubility measurements also result in small changes to calculated free energy penalties in the SI.

5. When discussing the binding stoichiometry of proSP-C BRICHOS, it should be mentioned that it under normal conditions exists as trimer and how this affect the determined binding stoichiometry (see e.g. Biverstal H, et al. Biochim Biophys Acta

1854, 835-843 (2015) or Willander H, et al. Proc Natl Acad Sci U S A 109, 2325-2329 (2012))

RESPONSE: For Fig 2: What is measured here is the total fluorescence of free vs fibril-bound Brichos. The trimeric state of free Brichos does not change this other than possible mild self-quenching. So the percentage of free Brichos could be underestimated a little, causing a modest overestimate of the Brichos binding stoichiometry. For Fig 5: What is measured here is the total fluorescence of free Brichos. Although self-quenching in trimers could artefactually reduce this, we use arbitrary units for the free Brichos concentration so the only potential effect on our fitting results is to change the arbitrary proportionality constants. There is therefore no effect on the fitted binding stoichiometry.

ACTION: We have added the effects of Brichos trimerisation on fitting of Figs 2 and 5 to the corresponding results sections (line 165-167, page 6 and line 309-312, page 12).

6. The discussion highlights the implication of the structural defects for AD pathology. What level of supersaturation is expected in vivo? Normal CSF concentrations of Aβ are clearly below the solubility limit, and the concentration decreases further during the disease. This would indicate that defects, and thereby secondary nucleation, may not have a major role in vivo.

RESPONSE: The chemical environments in CSF and in the brain interstitial fluid are very different to the chemical environment in vitro. It is therefore not expected that the solubility of Aβ40 with respect to fibrils in vitro and in vivo will be especially similar. Clearly, for it to be possible for Aβ40 fibrils to accumulate, the concentrations of Aβ40 must be above the local solubility limit. The extent of supersaturation in these environments is however not known. Nonetheless, even at low or no supersaturation, the laws of thermodynamics guarantee that defects still form, with stoichiometry governed by the Boltzmann distribution.

ACTION: We have rewritten the final Results section to now focus on proving that the defect site stoichiometry is still high enough in Aβ40 fibrils produced without supersaturation that most secondary nucleation still occurs at defects regardless of Aβ40 supersaturation levels. We have also added a new Discussion paragraph that points out that in vivo relevance must therefore be determined by factors other than supersaturation-level-control of defect stoichiometry.

Minor issues:

Page 7, line 197: should it be '< 90%' ?

RESPONSE+ACTION: We have now rewritten this sentence for greater clarity.

Page 16, line 483: change Kd to KD

RESPONSE+ACTION: We thank the reviewer for catching this. We have now done this.

Page 24, Ref 39: missing details

RESPONSE+ACTION: It is a master's thesis rather than a publication, therefore it has fewer details than might be expected. In keeping with citing conventions, we have now added the following details to the citation: university name, department name, and a url.

Page 26, Ref 55: missing details

RESPONSE+ACTION: We have now added the details

<https://doi.org/10.1039/D3SC06343G>

SI, page 4, line100: should it be ' $x = 2$ '?

RESPONSE+ACTION: Yes indeed, we thank the reviewer for catching this.

SI, Figure S3: What is the temperature for the NMR data? Both 60 and 37 deg are mentioned in the legend.

RESPONSE+ACTION: We have changed the legend to mention only 60 degrees, which is the correct temperature.

Reviewer #2 (Remarks to the Author):

In this manuscript the authors describe the mechanisms by which structural defects on amyloid- β fibrils act as secondary nucleation sites through analyzing binding kinetics of Brichos chaperon domain onto fibril surfaces. They also demonstrate that these defects and thereby secondary nucleation can be strongly inhibited by decreasing monomer concentration or by steadily decreasing reaction temperature. These so-called "annealed" A β fibrils are structurally indistinguishable from control fibrils and yet contain far fewer secondary nucleation sites and have much lower secondary nucleation propensity.

The manuscript is well-written and clear. Considering the recent approval of anti-A β antibodies for symptomatic treatment of Alzheimer's disease, the manuscript is quite timely and moves the field forward. No major issues in methodology or interpretation is evident, but a number of minor issues should be addressed:

RESPONSE: We are very grateful for the overall positive assessment of our manuscript.

1. Data shown in Fig. 2b should be further clarified. First, individual data points should be reported instead of mean \pm SD, considering that N=2 for the blue curve. Second, the accuracy of the data points for the blue curve needs to be double-checked for (i) Brichos concentrations tested for the two different conditions do not overlap, (ii) the concentrations tested for the "blue" condition are not well spaced (please pay particular attention to the 6th data point). Third, the authors should specify whether the regression line is calculated for the blue curve only.

RESPONSE (i): in Fig 2d (now 2e) this is an error: for blue the x values are free Brichos and for red they are total Brichos. In Fig 2b (now 2c) all data points use measured free Brichos as x values and therefore do not automatically overlap.

ACTION (i): We have replotted Fig 2b (now 2c) as individual data points, and correctly labelled the x axis as "Free Brichos concentration".

RESPONSE (ii): the apparently unusual spacing is not an error. It comes from three factors. First, we sampled more frequently at lower concentrations where the curvature is greater, to better constrain the regression to the Langmuir model. Second, the x values are free Brichos concentrations measured, not total Brichos concentrations tested, and so some apparent clustering is caused by random measurement errors. Third, data points 6 and 7 used relatively similar total Brichos concentrations. Also, if we include the red dataset in the regressions then the spacing becomes more visually even.

ACTION (ii): We have added to the Methods (Line 567-569) a note that more measurements were taken at lower total Brichos concentrations, to better constrain the fits to the Langmuir model.

RESPONSE (iii): the regression curve was calculated on the blue dataset only. Given the close similarity between datasets, and since our model does not predict stoichiometry or KD differences for the underlying methods, on reflection we think it is more appropriate to instead fit to both datasets globally.

ACTION (iii): We have re-performed the regression so that it is now applied to both red and blue datasets simultaneously, and stated this in the caption of Figure 2.

2. Also in Fig. 2b, for the 7th Brichos concentration tested, the higher of the two data points is likely above the maximum coverage value (based on the binding stoichiometry determined from the fit curve). The authors should explain the potential causes of this apparent measurement error. Also, the coefficients of determination (R^2) should be reported for curves in Fig 2 as was done in Fig. 5.

RESPONSE: This has happened because at this datapoint we are near the 10% detection limit of MDS, with only 12% of the fluorescent signal coming from bound Brichos according to the gaussian fitting. The re-fitting of the Langmuir model to all data rather than just the blue data has increased the fitted maximum coverage value just enough that the outlier is no longer above it. Nonetheless such data points still carry significant error.

ACTION: We have added the R^2 values to the caption. We have also added a new panel before Fig 2b in response to another reviewer's comment showing % bound vs total Brichos, which is what is actually directly inferred from the MDS measurements, and illustrated the detection limit on this. We explain the likely cause of measurement error in the outlier in Results 2.1 first paragraph.

3. In SI Section 1.2.2, for 2-monomer-thick fibril, x should be 2, not 4. Fortunately, free energy differences were calculated using $x = 2$; so, a textual correction will be sufficient.

RESPONSE+ACTION: We thank the reviewer for catching this typo; we will correct it. As the reviewer points out, the calculations were performed with the correct x value.

4. In SI Section 3, K_d should be K_D (equations 16 and 17) to be consistent with the rest of the manuscript. The same goes for line 483 in the main text.

5. In SI Fig. 3 caption, NMR spectra is reported to be obtained at 37°C, but the text and title of the legend suggest 60°C.

RESPONSE+ACTION (4&5): We thank the reviewer for catching these typos; we have corrected them both.

6. Several paragraphs of the Results section (starting in lines 283, 295, 375 and 407) rather belong to the Discussion section, some of which contain material that are repeated in the Discussion.

RESPONSE (line 283): This paragraph was written with a Discussion-like tone, but it is key to understanding the cryo-EM analysis that defects will not be detectable but morphological differences will.

ACTION: We have reduced this paragraph to two sentences and merged it into the preceding paragraph to make this point clear.

RESPONSE (line 295): This paragraph describes novel results presented graphically in the SI, although it is not written in a way that makes this clear.

ACTION: In response to another reviewer's comment we have merged the relevant SI figure into Fig 4. We have also rewritten this paragraph to remove any potential discussion-like tone.

RESPONSE (line 375): This paragraph uses a new although simple calculation to demonstrate an important new result that even without supersaturation secondary nucleation still occurs predominantly at growth defects.

ACTION: In response to another reviewer's comment we have heavily rewritten this Results section to focus on this proof. In response to this comment we have also attempted to remove any discussion-like tone from the paragraph.

RESPONSE (line 407): This paragraph could indeed be reasonably moved to the Discussion and we thank the reviewer for pointing this out.

ACTION: We ultimately decided to remove this paragraph due to its speculative nature and the need to satisfy journal length constraints.

7. Brichos should be adequately introduced and relevant reference(s) should be cited.

RESPONSE+ACTION: We have now introduced Brichos with suitable references at the end of the Introduction.

8. Fig. 1a could be cited at the end of the sentence starting in line 65.

RESPONSE+ACTION: We have now cited Fig. 1a at the end of this sentence.

Reviewer #3 (Remarks to the Author):

This study presents a compelling investigation into the role of structural defects in amyloid- β ($A\beta$) fibrils as catalysts for secondary nucleation, a key process in amyloid aggregation associated with neurodegenerative diseases. The authors employ an impressive combination of biophysical techniques including microfluidic diffusional sizing, fluorescence correlation spectroscopy, and cryo-EM, coupled with kinetic modeling, to demonstrate that secondary nucleation sites correspond to rare growth defects in fibrils. The findings significantly advance our understanding of amyloid aggregation mechanisms and have important implications for therapeutic strategies targeting secondary nucleation.

The study's major strengths lie in its novel mechanistic insights and methodological rigor. By quantitatively showing that annealed fibrils exhibit dramatically reduced secondary nucleation activity while maintaining standard morphology, the authors provide evidence linking defects to nucleation sites.

RESPONSE: We are again very grateful for the overall positive assessment of our manuscript.

However, I would like clarification from the authors on several important issues before recommending publication. They are listed below:

1. The demonstration that Brichos binds to these sites at substoichiometric ratios is based on accurate determination of [Stot]. However, no mention is made about the

labelling efficiency of the A488-Brichos sample and there is no discussion on whether this can significantly impact the conclusion.

RESPONSE: This is a good point. The Brichos concentrations we use refer to the concentration of labelled + unlabelled Brichos. Therefore, if the labelling efficiency is <100%, this will not affect the fitting results assuming labelled + unlabelled Brichos have similar binding thermodynamics.

ACTION: We have now measured the labelling efficiency using UV-Vis, finding it to be 90%. We have added these experiments as a new supplemental figure (S6) in a new SI section (SI Sec. 8). We briefly explain in the accompanying SI section that a lower labeling efficiency does not affect fitting results assuming labeled and unlabeled Brichos bind fibrils similarly. We now refer to this SI figure in the Methods section where Brichos labeling is introduced (line 530-531, page 18).

2. What do the Brichos concentration represent – are the reported values total protein concentrations or concentration of the labelled fractions?

RESPONSE: The Brichos concentration represents the total protein concentration (labeled+unlabeled Brichos).

ACTION: this is now stated explicitly in the Methods (line 532-533, page 18).

3. In Fig. 2D the fibril bound Brichos % increase with concentration but data is shown only up to 1.5 mM. It would be informative to see results at higher concentrations to assess saturation behavior.

RESPONSE: The saturation is already directly observable in the MDS data even without performing a regression although this is not clear from the way the data are plotted. The fraction of free vs bound Brichos was directly estimated using MDS, and this was later converted into concentrations. At the highest Brichos concentrations used, we are already at <10% of Brichos being bound. MDS cannot reliably distinguish 2 species when one is present at <10% of the concentration of the other. Therefore, using higher concentrations will not be able to demonstrate levels of saturation higher than this with any statistical significance.

ACTION: We have now inserted a new panel before panel 2b showing a plot of the % of Brichos that is free vs the total Brichos concentration, and showing that at the highest Brichos concentrations the bound Brichos % already drops below the detection limit. In the caption we clarify that the free vs bound brichos plot is then calculated from this.

4. The authors strangely use two different techniques for estimating Brichos binding to amyloid fibrils for experiments in Fig. 2 (MDS) and Fig. 5 (FCS). For proper comparison and consistency MDS and FCS should be used for both experiments.

RESPONSE: We used different techniques for Ab40 and Ab42 for historical reasons. Although comparison of Ab40 and Ab42 Brichos binding stoichiometry was never a key goal of the study, we nonetheless see the value in using both methods on one of the fibril types to check consistency between methods. We have therefore now also performed MDS on Ab40 fibrils in addition to the original FCS. (We have not included FCS experiments on Ab42 fibrils since this would take several more months for us to complete, and is not needed to demonstrate consistency now that MDS has been used on both fibril types.)

ACTION: We added the new MDS data on Ab40 as a new panel in Fig 5 and fitted globally for the stoichiometries along with the FCS data. We found similar stoichiometry values to

before when we solely used FCS data, although with increased accuracy. We now motivate the FCS data in the text (Results Sec. 2.4) as a confirmatory experiment to ensure that the different stoichiometries of annealed and control fibrils are not somehow an artefact of the MDS technique.

5. The "annealing" concept, while central to the study, isn't clearly defined until the Results section and would benefit from earlier introduction.

RESPONSE: We take the reviewer's point - currently Results section 3 is both too long and contains much material which is already known and better-placed in the Introduction.

ACTION: We have moved substantial material concerning defect stoichiometry and their removal through annealing from Results section 3 to the Introduction.

6. While data shown to infer Brichos binding to defect sites is compelling it can benefit from a more direct experimental evidence. For example recent work using single molecule TIRF measurements have directly captured secondary nucleation events from visualisation of branching

(<https://pubs.acs.org/doi/full/10.1021/jacs.1c07228>). These types of direct measurements could complement the results described here.

RESPONSE: Most plausible defect structures are expected to be too small and irregular to visualize at the resolution of TIRF or other microscopy techniques such as AFM or cryo-TEM. However, we in fact already found direct evidence of defects using cryo-TEM, via their nonlocal effects on fibril structure. We previously relegated these images to the SI, where they may have gone unnoticed.

ACTION: We have added the direct cryo-TEM images of defects to figure 4, and added text to the associated Results section to explain that defect stoichiometry can nonetheless not be reliably quantified by microscopy (line 268-273, page 11).

7. The study assumes all secondary nucleation sites as identical between annealed and control fibrils, differing only in frequency. This point needs clarification.

RESPONSE: In Results section 2.3 it is explicitly stated that defect structure may differ between annealed and control fibrils (lines 280-282 of the submitted manuscript). In section 2.5 (lines 371-374 of the submitted manuscript) it is explained that the kinetic and binding data implies there is little difference in structure: "The close correspondence between secondary nucleation site stoichiometry and secondary nucleation propensity implies that there is little if any structural difference between the secondary nucleation sites in annealed and control fibrils". No other important results or conclusions in the manuscript depend in turn on this conclusion, however.

ACTION: We have added some sentences to Results sections 2.5 and 2.6 to make this point clearer.

Reviewer #2 (Remarks to the Author):

The authors have revised the manuscript to answer to all my (minor) comments. The manuscript is well-organized and timely, and it moves the field forward. Publication is recommended.

Additional minor issues that arose during revision:

- In line 575, the centrifugation speed should be given in g-force not in rpm, since the latter depends on the instrument.

We have now written the centrifugation speed in g-force.

- In their rebuttal, the authors stated that they added R^2 values of the fit curves to Fig. 2 caption, but this is only true for the fit curve in panel C but not for the fit curves in panel E.

Panel E actually features exclusively model projections, not fitting. These models use parameters fitted in panels c and d. We have now added a mean-squared-error to panel d (in line with convention in amyloid kinetic curve fitting instead of r squared). We have also changed the caption for panel e to clarify that it contains only projections, not fits, and added r-squared values.

- In Fig. S5, panel B appears to have considerably better contrast than panel A. Can the authors improve the cryo-TEM image in panel A?

The contrast in figure S5a was already adjusted and represents the best that we were able to achieve.

- Considering that the solubility data reported in Tables S1 and S2 are generally consistent with each other, I see no clear reason for replacing Fig. 3d in the original manuscript with Fig. 3c in the revised manuscript (which has less data points and is less informative overall). The manuscript would improve if the authors replaced Fig. 3c with the original Fig. 3d and added 95% CI error bars (now reported in Table S2).

We have replaced Fig.3c with the original Fig.3d with 95% CI error bars.